# Learning Adversarial Low-rank Markov Decision Processes with Unknown Transition and Full-information Feedback

**Canzhe Zhao**
Shanghai Jiao Tong University
canzhezhao@sjtu.edu.cn

**Ruofeng Yang**
Shanghai Jiao Tong University
wanshuiyin@sjtu.edu.cn

**Baoxiang Wang**
The Chinese University of Hong Kong, Shenzhen
bxiangwang@cuhk.edu.cn

**Xuezhou Zhang**
Boston University
xuezhouz@bu.edu

**Shuai Li**[*]
Shanghai Jiao Tong University
shuaili8@sjtu.edu.cn

## Abstract

In this work, we study the low-rank MDPs with adversarially changed losses in the full-information feedback setting. In particular, the unknown transition probability kernel admits a low-rank matrix decomposition [Uehara et al., 2022], and the loss functions may change adversarially but are revealed to the learner at the end of each episode. We propose a policy optimization-based algorithm POLO, and we prove that it attains the $\widetilde{O}(K^{5/6}A^{1/2}d\ln(1+M)/(1-\gamma)^2)$ regret guarantee, where $d$ is rank of the transition kernel (and hence the dimension of the unknown representations), $A$ is the cardinality of the action space, $M$ is the cardinality of the model class that contains all the plausible representations, and $\gamma$ is the discounted factor. Notably, our algorithm is oracle-efficient and has a regret guarantee with no dependence on the size of potentially arbitrarily large state space. Furthermore, we also prove an $\Omega(\frac{\gamma^2}{1-\gamma}\sqrt{dAK})$ regret lower bound for this problem, showing that low-rank MDPs are statistically more difficult to learn than linear MDPs in the regret minimization setting. To the best of our knowledge, we present the first algorithm that interleaves representation learning, exploration, and exploitation to achieve the sublinear regret guarantee for RL with nonlinear function approximation and adversarial losses.

## 1 Introduction

In reinforcement learning (RL), the goal is to learn a (near) optimal policy through the interactions between the learner and the environment, which is typically modeled as the Markov decision processes (MDPs) [Feinberg, 1996]. When the state and action spaces are finite, many works have established the minimax (near) optimal regret guarantees for MDPs with finite horizon [Azar et al., 2017] and MDPs with infinite horizon [Tossou et al., 2019, He et al., 2021b]. In real applications of RL, however, the state and action spaces may be arbitrarily large and even infinite, which may lead to the curse of dimensionality. To tackle this issue, a common approach is *function approximation*, which

---

[*]Corresponding author.

37th Conference on Neural Information Processing Systems (NeurIPS 2023).

approximates the value functions of given policies with the leverage of feature mappings. Assuming that the feature mapping which embeds the state-action pairs to a low dimensional embedding space is known, RL with linear function approximation has been well studied recently. In particular, linear mixture MDPs [Ayoub et al., 2020] and linear MDPs [Jin et al., 2020b] are the two models of RL with linear function approximation that have been extensively studied. Notably, their (near) optimal regret guarantees are established by Zhou et al. [2021] and He et al. [2023] respectively. Nevertheless, in scenarios with complex and large-scale data, attaining the true underlying feature mappings might be unrealistic, and thus representation learning is needed. Many empirical works have shown that representation learning can accelerate the sample and computation efficiency of RL [Silver et al., 2018, Laskin et al., 2020, Yang and Nachum, 2021, Stooke et al., 2021, Schwarzer et al., 2021, Xie et al., 2022]. However, representation learning is provably more difficult in RL and other sequential decision-making problems than in non-sequential and non-interactive problems (*e.g.*, supervised learning) [Du et al., 2020, Wang et al., 2021a, Weisz et al., 2021, Uehara et al., 2022]. To pursue sample-efficient RL in the presence of representation learning, recent works have made initial attempts to study the theoretical guarantees of representation learning in RL under the fixed or stochastic loss functions [Uehara et al., 2022, Zhang et al., 2022].

In practice, however, it might be stringent to assume that the loss functions are stochastic. To tackle this issue, Even-Dar et al. [2009] and Yu et al. [2009] propose the first algorithms with provably theoretical guarantees that can handle adversarial MDPs, where the loss functions may change adversarially in each episode. Subsequently, most of the works in this line of research focus on learning tabular MDPs with adversarial loss functions [Neu et al., 2010a,b, 2012, Arora et al., 2012, Zimin and Neu, 2013, Dekel and Hazan, 2013, Dick et al., 2014, Rosenberg and Mansour, 2019a,b, Jin and Luo, 2020, Jin et al., 2020a, Shani et al., 2020b, Chen et al., 2021, Ghasemi et al., 2021, Rosenberg and Mansour, 2021, Jin et al., 2021b, Dai et al., 2022, Chen et al., 2022]. To learn adversarial MDPs with large state and action spaces, some recent works study RL with adversarial loss functions and linear function approximation [Cai et al., 2020, Neu and Olkhovskaya, 2021, Luo et al., 2021b,a, He et al., 2022, Zhao et al., 2023, Kong et al., 2023]. However, all these existing works assume that the state-action feature representations are known. As aforementioned, in complex and high-dimensional environments, the application of these algorithms may be still hindered due to the potential difficulty of knowing the true feature mappings a priori. Therefore, the following question naturally remains open:

*Can we devise an algorithm to simultaneously tackle the representation learning and adversarially changed loss functions in reinforcement learning?*

In this work, we give an affirmative answer to the above question in the setting of adversarial low-rank MDPs with full-information feedback. Specifically, in this problem, the unknown transition probability kernel admits a low-rank matrix decomposition but the true representations regarding the transitions are not known a priori. Meanwhile, the loss functions are arbitrarily chosen by an adversary across episodes and are revealed to the learner after each episode.

To solve this problem, we propose a policy optimization-based algorithm, which we call **P**olicy **O**ptimization for **LO**w-rank MDPs (POLO). Specifically, our POLO algorithm obtains an $\widetilde{O}(K^{5/6}A^{1/2}d\ln(1 + M)/(1 - \gamma)^2)$ regret guarantee for adversarial low-rank MDPs in the full-information feedback setting and is oracle-efficient. Algorithmically, POLO follows similar ideas of optimistic policy optimization methods in that it first constructs optimistic value function estimates and then runs online mirror descent (OMD) over the optimistic value estimates to deal with the adversarially changed loss functions [Shani et al., 2020b, Cai et al., 2020, He et al., 2022, Chen et al., 2022]. However, in the presence of representation learning, the exploration and exploitation needed to learn the adversarial MDPs are more difficult than them in the tabular case [Shani et al., 2020b, Chen et al., 2022] and in the linear case [Cai et al., 2020, He et al., 2022], where the exploration and exploitation can be essentially well balanced by directly modifying the algorithms studying stochastic MDPs in the tabular and linear cases. Concretely, to learn stochastic low-rank MDPs, previous works perform maximum likelihood estimation (MLE) over the experienced transitions [Agarwal et al., 2020, Uehara et al., 2022, Zhang et al., 2022]. Though the balance of representation learning, exploration, and exploitation can be simultaneously handled by the algorithms in these works, these algorithms intrinsically have no regret guarantees but only sample complexity guarantees even in the stochastic case, since these algorithms need to take actions uniformly at certain steps

Table 1: Comparisons of regret bounds with most related works studying adversarial RL with function approximation under unknown transitions. $K$ is the number of episodes, $d$ is the ambient dimension of the feature mapping, $\gamma$ is the discounted factor for infinite-horizon MDPs, and $S$, $A$, and $M$ are the cardinality of the state space, action space, and model class, respectively. Note that the dependence on $\gamma$ is not strictly comparable since some works originally studying finite-horizon MDPs and these results are translated into results for infinite-horizon MDPs by substituting horizon length $H$ with $\Theta(1/(1-\gamma))$. The column of "unknown features" indicates whether the algorithm can work in the case when no true feature mappings are known a priori.

| Algorithm | Model | Feedback | Regret | Unknown Features |
|---|---|---|---|---|
| OPPO [Cai et al., 2020] | Linear Mixture MDPs | Full-information | $\widetilde{O}\left(d\sqrt{K}/(1-\gamma)^2\right)$ | ✗ |
| POWERS [He et al., 2022] | Linear Mixture MDPs | Full-information | $\widetilde{O}\left(d\sqrt{K}/(1-\gamma)^{3/2}\right)$ | ✗ |
| LSUOB-REPS Zhao et al. [2023] | Linear Mixture MDPs | Bandit Feedback | $\widetilde{O}\left(dS^2\sqrt{K} + \sqrt{\frac{SAK}{(1-\gamma)}}\right)$ | ✗ |
| Luo et al. [2021a] | Linear MDPs | Bandit Feedback | $\widetilde{O}\left(d^2 K^{14/15}/(1-\gamma)^4\right)$ | ✗ |
| Dai et al. [2023] | Linear MDPs | Bandit Feedback | $\widetilde{O}\left(\frac{A^{1/9}d^{2/3}K^{8/9}}{(1-\gamma)^{20/9}}\right)$ | ✗ |
| PO-LSBE Sherman et al. [2023] | Linear MDPs | Bandit Feedback | $\widetilde{O}\left(\frac{dK^{6/7}}{(1-\gamma)^2} + \frac{d^{3/2}K^{5/7}}{(1-\gamma)^4}\right)$ | ✗ |
| GLAP Kong et al. [2023] | Linear MDPs | Bandit Feedback | $\widetilde{O}\left(d^{7/5}H^{12/5}K^{4/5}\right)$ | ✗ |
| OPPO+ Zhong and Zhang [2023] | Linear MDPs | Full-information | $\widetilde{O}\left(\frac{d^{3/4}K^{3/4}+d^{5/2}\sqrt{K}}{(1-\gamma)^2}\right)$ | ✗ |
| POLO (**Ours**) | Low-rank MDPs | Full-information | $\widetilde{O}\left(\frac{K^{5/6}A^{1/2}d\ln(1+M)}{(1-\gamma)^2}\right)$ $\Omega\left(\frac{\gamma^2}{1-\gamma}\sqrt{dAK}\right)$ | ✔ |

in each episode (*cf.*, Lemma 9 of Uehara et al. [2022]).[2] Hence, a straightforward adaption of their methods from the stochastic setting to the adversarial setting will fail to learn adversarial low-rank MDPs. To cope with this issue, we carefully devise an algorithm with a *doubled exploration and exploitation* scheme, which interleaves (a) the exploration over transitions required in representation learning; and (b) the exploration and exploitation suggested by the policy optimization. To this end, our algorithm adopts a mixed roll-out policy, which consists of a uniformly explorative policy and a policy optimized by OMD. Through carefully tuning the hyper-parameter of the mixing coefficient used in our mixed policy, we can avoid pulling actions uniformly at random to conduct exploration in each episode and only conduct uniform exploration at a certain fraction of all the episodes (see Section 3.1 for details). Besides, unlike tabular and linear (mixture) MDPs, it is in general hard to achieve the point-wise optimism for each state-action pair. Therefore, depart from previous methods [Shani et al., 2020b, Cai et al., 2020, He et al., 2022] conducting policy optimization in the *true* model (*i.e.*, the transition kernel characterized by the true representation), our algorithm conducts policy optimization in the fixed *learned* model with the epoch-based model update, which enables a new analysis scheme that only requires a *near optimism* at the initial state $s_0$ (see Section 3.2 for

---

[2]With the leverage of the common explore-then-commit (ETC) style conversion, the modified versions of these algorithms can obtain sublinear regret in the setting of low-rank MDPs with stochastic loss functions, but this conversion is still not able to deal with adversarial loss functions.

details). Also, we prove a regret lower bound of order $\Omega(\frac{\gamma^2}{1-\gamma}\sqrt{dAK})$ for low-rank MDPs with fixed loss functions, which thus also serves as a regret lower bound for our problem and indicates that low-rank MDPs are statistically more difficult to learn than linear MDPs in the regret minimization setting. To the best of our knowledge, this work makes the first step to establish an algorithm with a sublinear regret guarantee for adversarial low-rank MDPs, which permits RL with both nonlinear function approximation and adversarial loss functions. The concrete comparisons between the results of this work and those of previous works are summarized in Table 1.

## 1.1 Additional Related Works

**RL with Function Approximation**    Significant advances have emerged in RL with function approximation to cope with the curse of dimensionality in arbitrarily large state space or action space. In general, these results fall into two categories. The first category studies RL with linear function approximation, including linear MDPs [Yang and Wang, 2019, Jin et al., 2020b, Du et al., 2020, Zanette et al., 2020, Wang et al., 2020, 2021b, He et al., 2021a, Hu et al., 2022, He et al., 2023] and linear mixture MDPs [Ayoub et al., 2020, Zhang et al., 2021, Zhou et al., 2021, He et al., 2021a, Zhou and Gu, 2022, Wu et al., 2022, Min et al., 2022, Zhao et al., 2023]. Remarkably, He et al. [2023] and Zhou et al. [2021] obtain the nearly minimax optimal regret $\widetilde{O}(d\sqrt{H^3K})$ in linear MDPs and linear mixture MDPs respectively when the loss functions are fixed or stochastic. The other category studies RL with general function approximation. Amongst these works, Jiang et al. [2017], Dann et al. [2018], Sun et al. [2019], Du et al. [2019], and Jin et al. [2021a] study the MDPs satisfying the low Bellman-rank assumption, which assumes the Bellman error matrix has a low-rank factorization. Also, Du et al. [2021] consider a similar but slightly more general assumption termed as bounded bilinear rank. Besides, Russo and Roy [2013], Wang et al. [2020], Jin et al. [2021a], and Ishfaq et al. [2021] study low Eluder dimension assumption, which is originally proposed to characterize the complexity of function classes for bandit problems.

Representation learning in RL arises when the feature mapping that embeds the state-action pairs in RL with linear function approximation is no longer known a priori. Such a problem is typically studied in the setting of low-rank MDPs, which does not assume the feature mapping of state-action pairs is known. Consequently, the setting of low-rank MDPs strictly generalizes the setting of linear MDPs, but at the cost of being more difficult to learn due to potential nonlinear function approximation induced by representation learning. In this line of research, algorithms with provably sample complexity guarantees have been developed in both model-based methods [Agarwal et al., 2020, Ren et al., 2022, Uehara et al., 2022] and model-free methods [Modi et al., 2021, Zhang et al., 2022], respectively. The model-based algorithms of Agarwal et al. [2020], Ren et al. [2022] and Uehara et al. [2022] learn the representation from a given model class of transition probability kernels. In contrast, the model-free methods do not require model learning but may bear some limitations. In particular, Modi et al. [2021] assume the MDPs satisfying the minimal reachability assumption, and the sample complexity of the algorithm of Zhang et al. [2022] only holds for a special class of low-rank MDPs called block MDPs. Besides, representation learning in Markov games has also been investigated recently [Ni et al., 2022].

**RL with Adversarial Losses**    Recent years have witnessed significant advances in learning RL with adversarial losses in the tabular case [Neu et al., 2010a,b, 2012, Arora et al., 2012, Zimin and Neu, 2013, Dekel and Hazan, 2013, Dick et al., 2014, Rosenberg and Mansour, 2019a,b, Jin and Luo, 2020, Jin et al., 2020a, Shani et al., 2020b, Chen et al., 2021, Ghasemi et al., 2021, Rosenberg and Mansour, 2021, Jin et al., 2021b, Dai et al., 2022, Chen et al., 2022]. When it comes to the setting of linear function approximation, various policy optimization-based methods have been established to solve adversarial linear mixture MDPs [Cai et al., 2020, He et al., 2022] and adversarial linear MDPs [Luo et al., 2021a,b, Dai et al., 2023, Sherman et al., 2023, Zhong and Zhang, 2023]. The other line of works studies RL with linear function approximation and adversarial losses using occupancy measure-based methods [Neu and Olkhovskaya, 2021, Zhao et al., 2023, Kong et al., 2023]. To the best of our knowledge, however, there are no works in existing literature studying RL with both nonlinear function approximation and adversarial loss functions.

## 2 Preliminaries

We consider episodic infinite horizon low-rank MDPs with adversarial loss functions, the preliminaries of which are introduced as follows.

**Episodic Infinite-horizon Adversarial MDPs**   An episodic infinite horizon adversarial MDP is denoted by a tuple $(\mathcal{S}, \mathcal{A}, P^\star, \{\ell_k\}_{k=1}^K, \gamma, d_0)$,[3] where $\mathcal{S}$ is the state space (with potentially infinitely many states), $\mathcal{A}$ is the finite action space with cardinality $|\mathcal{A}| = A$, $P^\star : \mathcal{S} \times \mathcal{A} \times \mathcal{S} \to [0, 1]$ is the transition probability kernel such that $P^\star(s' \mid s, a)$ is the probability of transferring to state $s'$ from state $s$ after executing action $a$, $\gamma \in [0, 1)$ is the discount factor, $d_0 \in \Delta(\mathcal{S})$ is the initial distribution over the state space, and $\ell_k : \mathcal{S} \times \mathcal{A} \to [0, 1]$ is the loss function of episode $k$ chosen by the adversary. For the ease of exposition, we assume $d_0$ is known.

In this work, we consider a special class of MDPs called *low-rank MDPs* [Agarwal et al., 2020, Uehara et al., 2022, Zhang et al., 2022]. Specifically, instead of assuming a known true feature mapping, low-rank MDPs only assume that the transition probability kernel $P^\star$ admits a low-rank decomposition, with the formal definition given as follows.

**Definition 2.1** (Low-rank MDPs). *An MDP is a low-rank MDP if there exist two feature embedding functions $\phi^\star : \mathcal{S} \times \mathcal{A} \to \mathbb{R}^d$ and $\mu^\star : \mathcal{S} \to \mathbb{R}^d$ such that for any $(s, a, s') \in \mathcal{S} \times \mathcal{A} \times \mathcal{S}$, $P^\star(s' \mid s, a) = \mu^\star(s')^\top \phi^\star(s, a)$, where $\|\phi^\star(s, a)\|_2 \leq 1$ and for any function $g : \mathcal{S} \to [0, 1], \left\| \int \mu^\star(s) g(s) \mathrm{d}(s) \right\|_2 \leq \sqrt{d}$.*

Note that the regularity assumption imposed over $\phi^\star$ and $\mu^\star$ is only for the purpose of normalization.

**Function Approximation**   When the state space is arbitrarily large, function approximation is usually considered to permit sample-efficient learning for MDPs. Since the true feature mapping of state-action pairs is not known a priori in the low-rank MDPs, to make this problem tractable, we assume the access to a *realizable* model class as previous works [Agarwal et al., 2020, Uehara et al., 2022], detailed in the following.

**Assumption 2.1.** *There exists a known model class $\mathcal{M} = \{(\mu, \phi) : \mu \in \Psi, \phi \in \Phi\}$ satisfying (a) $\|\phi(s, a)\|_2 \leq 1$ and $\int \mu^\top(s') \phi(s, a) \mathrm{d}(s') = 1$ for any $(s, a, s') \in \mathcal{S} \times \mathcal{A} \times \mathcal{S}$, $\mu \in \Psi$, $\phi \in \Phi$; and (b) $\left\| \int \mu(s) g(s) \mathrm{d}(s) \right\|_2 \leq \sqrt{d}$ for any function $g : \mathcal{S} \to [0, 1]$. Moreover, it holds that $\mu^\star \in \Psi$ and $\phi^\star \in \Phi$.*

Throughout this paper, for the sake of brevity, we assume that the cardinality of $\Psi$ and $\Phi$ are finite, meaning that $\mathcal{M}$ also has bounded cardinality $M = |\mathcal{M}|$. However, note that extending the analyses to the function classes with infinite cardinality but bounded statistical complexity (*e.g.*, VC dimension) is not technically difficult.

**Interaction Protocol**   We now introduce the interaction protocol between the learner and the environment. To begin with, denote by $d_P^\pi(s, a) = (1 - \gamma) \sum_{\tau=0}^\infty \gamma^\tau d_{P,\tau}^\pi(s, a)$ the state-action occupancy distribution, where $d_{P,\tau}^\pi(s, a)$ is the probability of visiting $(s, a)$ at step $\tau$ under policy $\pi$ and transition $P$. With a slight abuse of notation, let $d_P^\pi(s) = \sum_{a \in \mathcal{A}} d_P^\pi(s, a)$ be the state occupancy distribution, denoting the probability of visiting state $s$ under $\pi$ and $P$.

Ahead of time, an MDP is decided by the environment, and only the state space $\mathcal{S}$ and the action space $\mathcal{A}$ are revealed to the learner. Meanwhile, the adversary secretly chooses $K$ loss functions $\{\ell_k\}_{k=1}^K$, each of which will be used in one episode. The interaction will proceed in $K$ episodes. At the beginning of episode $k$, the learner chooses a stochastic policy $\pi_k : \mathcal{S} \times \mathcal{A} \to [0, 1]$, where $\pi_k(a \mid s)$ is probability of taking $a$ at state $s$. Starting from an initial state $s_0 \sim d_0$, the learner repeatedly executes policy $\pi_k$ until reaching the termination. After episode $k$ is terminated, the learner observes a trajectory $\{(s_{k,\tau}, a_{k,\tau})\}_\tau$ as well as the loss function $\ell_k$.

For each episode $k$ and each state-action pair $(s, a)$, the state-action value $Q_k^\pi(s, a)$ is defined as $Q_k^\pi(s, a) = \mathbb{E}\left[ \sum_{\tau=0}^\infty \gamma^\tau \ell_k(s_{k,\tau}, a_{k,\tau}) \Big| \pi, P^\star, (s_{k,0}, a_{k,0}) = (s, a) \right]$. Also define $V_k^\pi(s) = \mathbb{E}_{a \sim \pi(\cdot|s)}[Q_k^\pi(s, a)]$ and $V_k^\pi = \mathbb{E}_{s_0 \sim d_0}[V_k^\pi(s_0)]$. The learning objective is to minimize the *pseudo regret* with respect to $\pi^\star$, defined as

$$\mathcal{R}_K = \mathbb{E}\left[ \sum_{k=1}^K \left( V_k^{\pi_k} - V_k^{\pi^\star} \right) \right],$$

---

[3]Though we focus on episodic infinite-horizon MDPs in this work, note that it is not technically difficult to extend the analyses in this work to the case of episodic finite-horizon MDPs.

where the expectation is taken over the potential randomness of the algorithm, $\pi^\star \in \arg\min_{\pi \in \Pi} \sum_{k=1}^{K} V_k^\pi$ is the fixed optimal policy in hindsight and $\Pi$ is the set of stochastic policies.

## 3 Algorithm

This section presents our POLO algorithm with the pseudocode illustrated in Algorithm 1. At a high level, POLO leverages a mixed roll-out policy to conduct doubled exploration and exploitation, *i.e.*, (a) the exploration over transitions required by representation learning; and (b) the exploration and exploitation over adversarially changed loss functions required by the policy optimization (Section 3.1). To deal with the issue that only the *near optimism* at the initial state $s_0$ is available in low-rank MDPs, POLO conducts policy optimization in fixed *learned* models with the epoch-based model update, which enables a new analysis scheme (Section 3.2).

### 3.1 Doubled Exploration and Exploitation

Let $\tilde{\pi}_k$ be the policy of episode $k$ computed by policy optimization (*cf.*, Eq. (3)). At the beginning of episode $k$, our algorithm first collects a state $s_k \sim d_{P^\star}^{\tilde{\pi}_k}$ (Line 6) by invoking a geometric sampling *roll-in* procedure [Kakade and Langford, 2002, Agarwal et al., 2021, Uehara et al., 2022]. Starting from an initial state $s_0 \sim d_0$, at each step $\tau$, this roll-in procedure will terminate and return state $s_\tau$ with probability $1 - \gamma$, and otherwise will take action $a_\tau \sim \tilde{\pi}_k(\cdot \mid s_\tau)$ and transfer to the next state $s_\tau \sim P^\star(\cdot \mid s_\tau, a_\tau)$.

Then our algorithm will further interact with the environment in successive two steps after collecting $s_k \sim d_{P^\star}^{\tilde{\pi}_k}$ (Line 7 - Line 11) like previous works studying low-rank MDPs [Agarwal et al., 2020, Uehara et al., 2022, Zhang et al., 2022]. One of the main differences between previous algorithms and ours lies in how to deal with exploration and exploitation when interacting with the environment. In the case of stochastic low-rank MDPs, the key in the analysis is to ensure the (near) optimism of the optimal policy $\pi^\star$ in the learned model, which essentially requires to bound the performance gap $\widehat{V}_k^{\pi^\star} - V_k^{\pi^\star}$ by relating it with the model error regarding $\pi^\star$, *i.e.*, $\mathbb{E}_{(s,a) \sim d_{P^\star}^{\pi^\star}}[\|\hat{P}_k(\cdot \mid s, a) - P^\star(\cdot \mid s, a)\|_1]$, where $\hat{P}_k$ is the MLE solution defined in Eq. (1). However, this model error is not directly controllable, as the algorithm does not know the optimal policy $\pi^\star$ and can not collect data following $\pi^\star$ to bound the model error. Fortunately, by empirical process theory [Geer, 2000, Zhang, 2006], the model error $\mathbb{E}_{(s,a) \sim \rho_k}[\|\hat{P}_k(\cdot \mid s, a) - P^\star(\cdot \mid s, a)\|_1]$ regarding the executed policies $\{\tilde{\pi}_i\}_{i=1}^{k}$ is bounded, where $\rho_k(s, a) = \frac{1}{k} \sum_{i=1}^{k} d_{P^\star}^{\tilde{\pi}_i}(s, a)$. Therefore, previous works [Agarwal et al., 2020, Uehara et al., 2022, Zhang et al., 2022] ensure the optimism of the optimal policy $\pi^\star$ by applying importance weighting to change the measure from $d_{P^\star}^{\pi^\star}$ to $\rho_k$ when bounding the model error. The complication is that to control the ratio $\pi^\star(a \mid s)/\tilde{\pi}_i(a \mid s)$ for any $(s, a) \in \mathcal{S} \times \mathcal{A}$ and $i \in [K]$ in importance weighting, the algorithms in previous works take actions from the uniform distribution $U(\mathcal{A})$ over action space $\mathcal{A}$, which intuitively can be seen as conducting exploration over transitions required by representation learning. Consequently, though these algorithms enjoy excellent sample complexities, they intrinsically do not have regret guarantees due to the uniform exploration over action space, even in the stochastic setting. Moreover, to learn adversarial low-rank MDPs, it is required to take actions adaptively according to the observed loss functions in previous episodes instead of uniformly taking actions. To address this "conflict" so as to learn adversarial low-rank MDPs, we propose to use a mixed roll-out policy to interleave (a) the exploration over transitions required by representation learning; and (b) the exploration and exploitation over the adversarial loss functions by policy optimization, which we call *doubled exploration and exploitation* and is pivotal to achieving our regret bound as we will shortly see. Formally, our algorithm will conduct the exploration over the transitions with probability $\xi$ and execute policy $\tilde{\pi}_k$ optimized by OMD with probability $1 - \xi$ respectively, as shown in Line 7 - Line 11.

After interacting with the environment, the newly collected data in these two steps will be used to update the datasets (Line 14), and the empirical transition $\widehat{P}_k$ will be updated by performing MLE over the updated datasets by solving (Line 16)

$$\left(\widehat{\mu}_k, \widehat{\phi}_k\right) = \underset{(\mu, \phi) \in \mathcal{M}}{\arg\max} \, \mathbb{E}_{\mathcal{D}_k \cup \mathcal{D}_k'}\left[\ln \mu^\top(s')\phi(s, a)\right], \tag{1}$$

where we denote $\mathbb{E}_{\mathcal{D}}[f(s, a, s')] = \frac{1}{|\mathcal{D}|} \sum_{(s,a,s') \in \mathcal{D}} f(s, a, s')$.

---

**Algorithm 1** Policy Optimization for Low-rank MDPs (POLO)

---

1: **Input:** Mixing coefficient $\xi$, epoch length $L$, regularization coefficients $\{\lambda_k\}_{k=1}^K$, bonus coefficients $\{\alpha_k\}_{k=1}^K$, model class $\mathcal{M}$, number of episodes $K$, learning rate $\eta$.
2: **Initialization:** Set $\mathcal{D}_0 = \emptyset, \mathcal{D}'_0 = \emptyset$.
3: **for** $i = 1, 2, \ldots, \lceil K/L \rceil$ **do**
4:     Set $k_i = (i-1)L + 1$ and $\tilde{\pi}_{k_i}(\cdot \mid s)$ to be uniform for any $s \in \mathcal{S}$.
5:     **for** $k = k_i, k_i + 1, \ldots, k_i + L - 1$ **do**
6:         Sample $s_k$ from $d_{P^\star}^{\tilde{\pi}_k}$.
7:         Sample $c_k \sim \text{Ber}(1 - \xi)$.
8:         **if** $c_k = 1$ **then**
9:             Sample $a_k \sim \tilde{\pi}_k(\cdot \mid s_k), s'_k \sim P^\star(\cdot \mid s_k, a_k), a'_k \sim \tilde{\pi}_k(\cdot \mid s'_k), s''_k \sim P^\star(\cdot \mid s'_k, a'_k)$.
10:         **else**
11:             Sample $a_k \sim U(\mathcal{A}), s'_k \sim P^\star(\cdot \mid s_k, a_k), a'_k \sim U(\mathcal{A}), s''_k \sim P^\star(\cdot \mid s'_k, a'_k)$.
12:         **end if**
13:         Observe the loss function $\ell_k$.
14:         Update datasets $\mathcal{D}_k = \mathcal{D}_{k-1} \cup \{(s_k, a_k, s'_k)\}, \mathcal{D}'_k = \mathcal{D}'_{k-1} \cup \{(s'_k, a'_k, s''_k)\}$.
15:         **if** $k = k_i$ **then**
16:             Set the empirical transition $\widehat{P}_k(s' \mid s, a) = \widehat{\mu}_k(s')^\top \widehat{\phi}_k(s, a), \forall (s, a, s') \in \mathcal{S} \times \mathcal{A} \times \mathcal{S}$, via solving Eq. (1).
17:             Update the empirical covariance matrix $\widehat{\Sigma}_k = \sum_{(s,a) \in \mathcal{D}_k} \widehat{\phi}_k(s, a) \widehat{\phi}_k(s, a)^\top + \lambda_k I$.
18:             Set the bonus function $\widehat{b}_k(s, a) := \min(\alpha_k \|\widehat{\phi}_k(s, a)\|_{\widehat{\Sigma}_k^{-1}}, 2)/(1 - \gamma), \forall (s, a) \in \mathcal{S} \times \mathcal{A}$.
19:         **else**
20:             Set the empirical transition $\widehat{P}_k = \widehat{P}_{k_i}$ and bonus function $\widehat{b}_k = \widehat{b}_{k_i}$.
21:         **end if**
22:         Compute $\widehat{Q}_k^{\tilde{\pi}_k}(\cdot, \cdot) = \text{Policy-Evaluation}(\widehat{P}_k, \ell_k - \widehat{b}_k, \tilde{\pi}_k)$.
23:         Update policy $\tilde{\pi}_{k+1}(\cdot \mid \cdot) \propto \tilde{\pi}_k(\cdot \mid \cdot) \exp(-\eta \widehat{Q}_k^{\tilde{\pi}_k}(\cdot, \cdot))$.
24:     **end for**
25: **end for**

---

## 3.2 Policy Optimization in Fixed Learned Models

It remains to compute the policy $\tilde{\pi}_{k+1}$ to be used in the next episode. To this end, we resort to the canonical OMD framework to perform policy optimization like previous methods [Shani et al., 2020a, Cai et al., 2020, He et al., 2022]. However, previous OMD-based policy optimization methods for tabular and linear (mixture) MDPs [Shani et al., 2020a, Cai et al., 2020, He et al., 2022] critically depend on the point-wise optimism for each state-action pair, *i.e.*, $\widehat{Q}_k^{\tilde{\pi}_k}(s, a) \leq \ell_k(s, a) + \gamma[P^\star \widehat{V}_k^{\tilde{\pi}_k}](s, a)$, to enable the decomposition (*cf.*, Lemma 1 by Shani et al. [2020a])

$$\widehat{V}_k^{\tilde{\pi}_k}(s_0) - V_k^{\pi^\star}(s_0) = \mathbb{E}\left[\sum_{\tau=0}^\infty \gamma^\tau \left\langle \tilde{\pi}_k(\cdot \mid s_\tau) - \pi^\star(\cdot \mid s_\tau), \widehat{Q}_k^{\tilde{\pi}_k}(s_\tau, \cdot) \right\rangle \middle| \pi^\star, P^\star, s_0 \right]$$
$$+ \mathbb{E}\left[\sum_{\tau=0}^\infty \gamma^\tau \left( \widehat{Q}_k^{\tilde{\pi}_k}(s_\tau, a_\tau) - \ell_k(s_\tau, a_\tau) - \gamma \left[ P^\star \widehat{V}_k^{\tilde{\pi}_k} \right](s_\tau, a_\tau) \right) \middle| \pi^\star, P^\star, s_0 \right],$$

where $\widehat{V}_k^{\tilde{\pi}_k}$ and $\widehat{Q}_k^{\tilde{\pi}_k}$ are the state value and state-action value functions of $\tilde{\pi}_k$ on $(\widehat{P}_k, \ell_k - \widehat{b}_k)$ with $\widehat{b}_k$ as some bonus function and the expectation is taken over the randomness of sampling $a_\tau \sim \pi^\star(\cdot \mid s_\tau)$ and $s_{\tau+1} \sim P^\star(\cdot \mid s_\tau, a_\tau)$. The summation of the first term in the above display is contributed by competing with the optimal policy $\pi^\star$ in the *true* model $P^\star$ and can be bounded by common OMD analysis, which thus can be regarded as conducting policy optimization in the *true* model. The point-wise optimism guarantees that the second term is less than or equal to $0$.

Nevertheless, in low-rank MDPs, due to the unknown representation, it is generally hard to obtain the above point-wise optimism, which leaves the second optimism term unbounded. To cope with this issue, we instead consider the following decomposition:

$$\widehat{V}_k^{\tilde{\pi}_k}(s_0) - V_k^{\pi^\star}(s_0)$$

$$=\widehat{V}_k^{\tilde{\pi}_k}(s_0) - \widehat{V}_k^{\pi^\star}(s_0) + \widehat{V}_k^{\pi^\star}(s_0) - V_k^{\pi^\star}(s_0)$$

$$=\mathbb{E}\left[\sum_{\tau=0}^{\infty}\gamma^\tau\left\langle\tilde{\pi}_k(\cdot\mid s_{k,\tau}) - \pi^\star(\cdot\mid s_{k,\tau}), \widehat{Q}_k^{\tilde{\pi}_k}(s_{k,\tau},\cdot)\right\rangle\,\middle|\,\pi^\star, \widehat{P}_k, s_0\right] + \widehat{V}_k^{\pi^\star}(s_0) - V_k^{\pi^\star}(s_0), \quad (2)$$

where the first term is contributed by competing against the optimal policy $\pi^\star$ in the *learned* model $\widehat{P}_k$ and can be seen as conducting policy optimization in *learned* models. This decomposition will be amenable as long as we can achieve a *near optimism* at the initial state $s_0$, *i.e.*, $\widehat{V}_k^{\pi^\star}(s_0) - V_k^{\pi^\star}(s_0) \lesssim 0$, which turns out to be feasible for low-rank MDPs [Uehara et al., 2022]. However, there remains one more caveat. The first term in Eq. (2) is now no longer directly bounded by OMD analysis, due to the local update nature of OMD-based policy optimization at each state and the state occupancy distribution $d_{\widehat{P}_k}^{\pi^\star}$ now varies across different episodes. To address this issue, Algorithm 1 adopts an epoch-based transition update, in which one epoch has $L$ episodes and the model is only updated at the first episode in one epoch (Line 15 - Line 20).[4] Concretely, Algorithm 1 sets $\widehat{P}_k = \widehat{P}_{k_i}$ and $\widehat{b}_k = \widehat{b}_{k_i}$, where $k_i$ is the first episode of the epoch to which the episode $k$ belongs. In this manner, the learned model is *fixed* in one epoch, and thus the regret of dealing with the adversarial loss functions by competing against the optimal policy $\pi^\star$ can be bounded in one epoch. Subsequently, at the end of episode $k$, our algorithm first computes the optimistic value estimate $\widehat{Q}_k^{\tilde{\pi}_k}$ for current policy $\tilde{\pi}_k$ under $\widehat{P}_k$ together with the bonus-enhanced loss functions $\ell_k - \widehat{b}_k$ by policy evaluation (Line 22). Note that this boils down to planning in the setting of linear MDPs for given features in the learned model and this can be done computationally efficiently [Jin et al., 2020b]. Then the policy is updated by solving

$$\tilde{\pi}_{k+1}(\cdot\mid s) \in \operatorname*{arg\,min}_{\pi(\cdot\mid s)\in\Delta(\mathcal{A})} \eta\left\langle\pi(\cdot\mid s), \widehat{Q}_k^{\tilde{\pi}_k}(s,\cdot)\right\rangle + D_F(\pi(\cdot\mid s), \tilde{\pi}_k(\cdot\mid s)), \quad (3)$$

where $\eta > 0$ is the learning rate to be tuned later and $D_F(x,y) = F(x) - F(y) - \langle x-y, \nabla F(y)\rangle$ is the Bregman divergence induced by the regularizer $F$. With $F(\pi(\cdot\mid s)) = \sum_{a\in\mathcal{A}}\pi(a\mid s)\ln\pi(a\mid s)$ as the negative entropy, the closed-form solution to the above display is shown in Line 23, which can be regarded as a kind of soft policy improvement.

## 4  Analysis

### 4.1  Regret Upper Bound

The regret upper bound of our POLO algorithm for learning adversarial low-rank MDPs is guaranteed by the following theorem.

**Theorem 4.1.** *Suppose $K > d^6 A^3/(1-\gamma)^6$. For any adversarial low-rank MDP satisfying Definition 2.1, by setting the epoch length $L = K^{1/2}A^{-1/2}d^{-1}\xi(1-\gamma)$, learning rate $\eta = (1-\gamma)\sqrt{\ln A/(2L)}$, bonus coefficient $\alpha_k = O(\sqrt{\gamma(A/\xi + d^2)\ln(Mk/\delta)})$, regularization coefficient $\lambda_k = O(d\ln(Mk/\delta))$, mixing coefficient $\xi = K^{-1/6}A^{1/2}d/(1-\gamma)$, and $\delta = 1/K$, then the regret of Algorithm 1 is upper bounded by*

$$\mathcal{R}_K = O\left(\frac{K^{\frac{5}{6}}A^{\frac{1}{2}}d\ln\left(1 + AMK^2\right)}{(1-\gamma)^2}\right).$$

**Remark 4.1.** *Ignoring the dependence on all logarithmic factors but $M$, the regret upper bound can be simplified as $\widetilde{O}(K^{5/6}A^{1/2}d\ln(1+M)/(1-\gamma)^2)$. Comparing with the regret lower bound $\Omega(\frac{\gamma^2}{1-\gamma}\sqrt{dAK})$ in Theorem 4.2, the regret upper bound in Theorem 4.1 matches in $A$ up to a logarithmic factor but looses in factors of $K$ and $d$. Also, note that when $K$ is large enough such that $\xi$ and $L$ can be chosen as $\xi = K^{-1/6}A^{1/3}d^{2/3}/(1-\gamma)$ and $L = K^{1/2}A^{-1}d^{-2}\xi(1-\gamma) = K^{1/3}A^{-2/3}d^{-4/3} \geq 1$, meaning that $K \geq d^4 A^2$, the regret upper bound can be further optimized to $\widetilde{O}(K^{5/6}A^{1/3}d^{2/3}\ln(1+M)/(1-\gamma)^2)$. Note that this does not conflict with the regret lower bound in Section 4.3 since the magnitude of this upper bound is still larger than that of the regret lower bound as long as $K \geq A^{1/2}d^{-1/2}\gamma^6(1-\gamma)^3$.*

---

[4]Throughout this paper, we suppose for simplicity that the number of episodes $K$ is divisible by the epoch length $L$ considered.

## 4.2 Proof of Regret Upper Bound

We now present the proof of Theorem 4.1. To begin with, recall that in each episode $k$, after state $s_k$ is sampled from $d_{P^\star}^{\tilde{\pi}_k}$, the actual roll-out policy will be $\pi_k(\cdot \mid s) = \xi \cdot U(\mathcal{A}) + (1 - \xi) \cdot \tilde{\pi}_k(\cdot \mid s)$. Therefore, it holds that

$$
\begin{aligned}
\mathcal{R}_K &= \mathbb{E}\left[\sum_{k=1}^{K}\left(V_k^{\pi_k} - V_k^{\pi^\star}\right)\right] \\
&= \mathbb{E}\left[\sum_{k=1}^{K}\mathbb{I}\{c_k = 1\}\left(V_k^{\pi_k} - V_k^{\pi^\star}\right) + \mathbb{I}\{c_k = 0\}\left(V_k^{\pi_k} - V_k^{\pi^\star}\right)\right] \\
&\le \mathbb{E}\left[\sum_{k=1}^{K}\left(V_k^{\tilde{\pi}_k} - V_k^{\pi^\star}\right)\right] + \frac{\xi K}{(1-\gamma)},
\end{aligned}
\tag{4}
$$

where the inequality is due to that $\sum_{\tau=0}^{\infty}\gamma^\tau \ell_k(s_\tau, a_\tau) \in [0, 1/(1-\gamma)]$ holds for any episode $k$ and any trajectory $\{(s_\tau, a_\tau)\}_{\tau=0}^{\infty}$. We now turn to bound the first term in Eq. (4) by decomposing it into the following three terms

$$
\mathbb{E}\left[\sum_{k=1}^{K}\left(V_k^{\tilde{\pi}_k} - V_k^{\pi^\star}\right)\right] = \mathbb{E}\Bigg[\underbrace{\sum_{k=1}^{K}\left(V_k^{\tilde{\pi}_k} - \widehat{V}_k^{\tilde{\pi}_k}\right)}_{\text{ESTIMATION BIAS TERM}} + \underbrace{\sum_{k=1}^{K}\left(\widehat{V}_k^{\tilde{\pi}_k} - \widehat{V}_k^{\pi^\star}\right)}_{\text{OMD REGRET TERM}} + \underbrace{\sum_{k=1}^{K}\left(\widehat{V}_k^{\pi^\star} - V_k^{\pi^\star}\right)}_{\text{OPTIMISM TERM}}\Bigg].
\tag{5}
$$

**Bounding OMD REGRET TERM** The OMD REGRET TERM is contributed by competing against $\pi^\star$ using $\tilde{\pi}_k$ with $\widehat{Q}_k^{\tilde{\pi}_k}$ as loss function in the learned model $\widehat{P}_{k_i}$. This term is thus bounded by standard OMD analysis, detailed in the following lemma.

**Lemma 4.1.** *By setting* $\eta = (1-\gamma)\sqrt{\ln A/(2L)}$, *the* OMD REGRET TERM *is bounded as* $\mathbb{E}\left[\sum_{k=1}^{K}\left(\widehat{V}_k^{\tilde{\pi}_k} - \widehat{V}_k^{\pi^\star}\right)\right] \le \frac{K\sqrt{2\ln A}}{\sqrt{L}(1-\gamma)^2}$.

**Bounding OPTIMISM TERM** The OPTIMISM TERM is controlled by choosing appropriate bonus coefficient $\alpha_k$. Note that different from tabular and linear cases, the bonus functions and coefficients here are not devised to control the optimism for each state-action pair. Instead, they are devised to provide a (near) optimism only at the initial state $s_0$.

**Lemma 4.2.** *By setting* $\alpha_k = O(\sqrt{\gamma(A/\xi + d^2)\ln(Mk/\delta)})$ *and* $\lambda_k = O(d\ln(Mk/\delta))$, *with probability* $1 - \delta$, *the* OPTIMISM TERM *is bounded as* $\sum_{k=1}^{K}\left(\widehat{V}_k^{\pi^\star} - V_k^{\pi^\star}\right) \le (L + \sqrt{K})\sqrt{\frac{A\ln(MN/\delta)}{\xi(1-\gamma)^3}}$.

**Bounding ESTIMATION BIAS TERM** It remains to bound the ESTIMATION BIAS TERM, which comes from the difference between the values of running the same policy $\tilde{\pi}_k$ in the true model (*i.e.*, $P^\star$ and $\ell_k$) and the learned empirical model (*i.e.*, $\widehat{P}_k$ and $\ell_k - \widehat{b}_k$), respectively. This term can be translated into the error between the true model and the learned model using the common simulation lemma, which is thus bounded by the summation of bonus functions. Note that since the empirical features used to construct our bonus functions vary in each episode, we first relate the bonus functions with the fixed true feature $\phi^\star$ using the one-step trick [Uehara et al., 2022, Zhang et al., 2022], and finally bound this term with the leverage of the canonical elliptical potential lemma. The result is shown in the following lemma.

**Lemma 4.3.** *By setting* $\alpha_k = O(\sqrt{\gamma(A/\xi + d^2)\ln(Mk/\delta)})$ *and* $\lambda_k = O(d\ln(Mk/\delta))$, *with probability* $1 - \delta$, *the* ESTIMATION BIAS TERM *is bounded as* $\sum_{k=1}^{K}\left(V_k^{\tilde{\pi}_k} - \widehat{V}_k^{\tilde{\pi}_k}\right) \le O\left(\frac{d^2 A\sqrt{KL}}{\xi(1-\gamma)^3}\sqrt{\ln(1+K)\ln(MK/\delta)}\right)$.

We refer the readers to Appendix A for the proof of the above lemmas. The proof of Theorem 4.1 is now concluded by first combining Eq. (4), Eq. (5), Lemma 4.1, 4.2, and 4.3 and then choosing $L = K^{1/2}A^{-1/2}d^{-1}\xi(1-\gamma), \xi = K^{-1/6}A^{1/2}d/(1-\gamma)$, and $\delta = 1/K$.

Intuitively, the epoch length $L$ illustrates a trade-off between dealing with the adversarial losses and the representation learning over the unknown transitions. When $L$ is large, there will be fewer restarts in the running of OMD and thus the learner will suffer less regret contributed by dealing with the adversarial losses as shown by Lemma 4.1. In contrast, a smaller $L$ enables more frequent model updates, which leads to more accurate model estimation and less regret contributed by the representation learning as shown by Lemma 4.2 and 4.3.

### 4.3 Regret Lower Bound

This section presents the regret lower bound for learning adversarial low-rank MDPs with fixed loss functions in Theorem 4.2, which thus also serves as a regret lower bound for learning adversarial low-rank MDPs with full-information feedback.

**Theorem 4.2.** *Suppose $d \geq 8$, $S \geq d + 1$, $A \geq d - 3$, and $K \geq 2(d-4)A$. Then for any algorithm* Alg*, there exists an episodic infinite-horizon low-rank MDP $\mathcal{M}_{\text{Alg}}$ with fixed loss function such that the regret for this MDP is lower bounded by $\Omega(\frac{\gamma^2}{1-\gamma}\sqrt{dAK})$.*

*Proof Sketch.* At a high level, we construct $dA$ hard-to-learn low-rank MDP instances, which are difficult to distinguish in KL divergence but have very different optimal policies. In particular, all the constructed low-rank MDP instances have three levels of states, in which the only state in the first level is a fixed initial state and the states in the third level are absorbing states. Moreover, only one unique absorbing state in the third level has the lowest loss, which is termed as the "good state". In the constructed low-rank MDP instance $\mathcal{M}_{(i^\star, a^\star)}$, the learner can only take specific action to transfer to state $s_{2,i^\star}$ in the second level and then take the other specific action to transfer to the unique good state. Due to the unknown representations of state-action pairs, the learner needs to distinguish all these $dA$ low-rank MDP instances, which is essentially equivalent to dealing with a bandit problem with $dA$ "arms". The detailed proof of Theorem 4.2 is postponed to Appendix B. $\square$

**Remark 4.2.** *Theorem 4.2, to the best of our knowledge, provides the first regret lower bound for learning low-rank MDPs with fixed loss functions. We note that this regret lower bound can hold when $d \ll S$ and $d \ll A$, which thus means that this lower bound is non-trivial. Besides, the regret upper bound in our Theorem 4.1 matches the regret lower bound in $A$ up to a logarithmic factor but looses a factor of $\widetilde{O}(K^{1/3}d^{1/2}/((1-\gamma)\gamma^2))$. Importantly, compared with the regret upper bound $\widetilde{O}(d\sqrt{K/(1-\gamma)^3})$ of linear MDPs [He et al., 2023] (the finite horizon $H$ is substituted by the effective horizon $\Theta(1/(1-\gamma))$ in our infinite-horizon setting for a fair comparison), the dependence on $A$ in the regret lower bound of low-rank MDP shows a clear separation between low-rank MDPs and linear MDPs, which demonstrates that low-rank MDPs are statistically more difficult to learn than linear MDPs in the regret minimization setting.*

## 5 Conclusions

In this work, we study learning adversarial low-rank MDPs with unknown transition and full-information feedback. We prove that our proposed algorithm POLO achieves the $\widetilde{O}(K^{5/6}A^{1/2}d\ln(1+M)/(1-\gamma)^2)$ regret, which is the first sublinear regret guarantee for this challenging problem. The design of our proposed algorithm features (a) a doubled exploration and exploitation scheme to simultaneously learn the transitions and adversarial loss functions; and (b) policy optimization in the fixed learned models with epoch-based model update to enable a new analysis scheme that only requires the near optimism at the initial state instead of the point-wise optimism. Also, we prove an $\Omega(\frac{\gamma^2}{1-\gamma}\sqrt{dAK})$ regret lower bound for this problem, serving as the first regret lower bound for learning low-rank MDPs in the regret minimization setting. Besides, there also remain several interesting future directions to be explored. One natural question is whether it is possible to further optimize the dependence of our regret guarantee on the number of episodes $K$. The other question is how to also learn adversarial low-rank MDPs with only the bandit feedback available. This is also challenging since the current occupancy measure-based methods and policy optimization-based methods tackling adversarial MDPs with bandit feedback both depend on the point-wise optimism provided by the true feature mapping, which seems not feasible in low-rank MDPs. We hope our results may shed light on better understandings of RL with both nonlinear function approximation and adversarial losses and we leave the above extensions as our future works.

## Limitations

We note that in general our algorithm is oracle-efficient (given access to the MLE computation oracle in Eq. (1)) but may not be computationally efficient as previous works studying low-rank MDPs [Agarwal et al., 2020, Uehara et al., 2022, Zhang et al., 2022, Ni et al., 2022]. However, we also remark that in practice, these algorithms including ours are computationally feasible since the computation of MLE is only a standard supervised learning problem and can be implemented using gradient descent methods. The other limitation is that throughout this paper, we assume model class $\mathcal{M}$ with bounded cardinality $M$ and the regret upper bound of our algorithm has a logarithmic dependence on $M$. We remark that similar assumptions and dependence have also appeared in previous theoretical works studying RL with general function approximation [Jiang et al., 2017, Sun et al., 2019]. Also, extending the analyses to an infinite hypothesis class is possible if the hypothesis class has bounded statistical complexity [Agarwal et al., 2020].

## Acknowledgement

The corresponding author Shuai Li is supported by National Key Research and Development Program of China (2022ZD0114804) and National Natural Science Foundation of China (62376154, 62006151, 62076161). Baoxiang Wang is partially supported by National Natural Science Foundation of China (62106213, 72150002) and Shenzhen Science and Technology Program (RCBS20210609104356063, JCYJ20210324120011032).

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
