# Appendix

# A  Omitted Analysis of The Regret Upper Bound

In this section, we first introduce some further notations and then present the detailed analysis of Theorem 4.1. For the proofs in this section, we assume that $K$ is divisible by the epoch length $L$ considered for simplicity.

To begin with, denote by $\rho_k(s) = 1/k \sum_{i=1}^{k} d_{P^\star}^{\tilde{\pi}_i}(s)$ the averaged state occupancy distribution under $\{\tilde{\pi}_i\}_{i=1}^{k}$ and $P^\star$. Analogously, let $\rho'_k(s') = \sum_{(s,a)\in\mathcal{S}\times\mathcal{A}} \rho_k(s)\bar{\pi}_k(a \mid s)P^\star(s' \mid s,a)$ be the average next-state occupancy distribution after $s$ is sampled from $d_{P^\star}^{\tilde{\pi}_i}$, where $\bar{\pi}_k(\cdot \mid s) = \xi \cdot U(\mathcal{A}) + (1-\xi) \cdot 1/k \sum_{i=1}^{k} \tilde{\pi}_i(\cdot \mid s)$ is the averaged mixed roll-out policy. Define $f_k(s,a) = \left\| \widehat{P}_k(\cdot \mid s,a) - P^\star(\cdot \mid s,a) \right\|_1$ as the $\ell_1$-error between the estimate transition kernel $\widehat{P}_k$ and $P^\star$. Further, we define the following feature covariance matrices:

- $\widehat{\Sigma}_{k,\phi} = k\mathbb{E}_{(s,a)\sim\mathcal{D}_k} \left[ \phi(s,a)\phi(s,a)^\top \right] + \lambda_k I,$
- $\Sigma_{\rho_k,\phi} = k\mathbb{E}_{(s,a)\sim\rho_k} \left[ \phi(s,a)\phi(s,a)^\top \right] + \lambda_k I,$
- $\Sigma_{\rho_k\times\bar{\pi}_k,\phi} = k\mathbb{E}_{s\sim\rho_k,a\sim\bar{\pi}_k} \left[ \phi(s,a)\phi(s,a)^\top \right] + \lambda_k I.$

For notational convenience, we abbreviate $\widehat{\Sigma}_{k,\phi_k}$ as $\widehat{\Sigma}_k^{-1}$. Also, note that $\widehat{\Sigma}_{k,\phi}$ is an unbiased estimate of $\Sigma_{\rho_k\times\bar{\pi}_k,\phi}$.

## A.1  Bounding OMD REGRET TERM

We first present the proof of Lemma 4.1, which follows from the standard OMD analysis.

*Proof of Lemma 4.1.* We first consider fixed initial state $s_0$. For some fixed $i \in [N]$, one can see that the OMD regret in episode $k \in \{k_i, k_i + 1, \ldots, k_i + L - 1\}$ can be written recursively as

$$
\begin{aligned}
&\widehat{V}_k^{\tilde{\pi}_k}(s_0) - \widehat{V}_k^{\pi^\star}(s_0) \\
&= \left\langle \tilde{\pi}_k(\cdot \mid s_0), \widehat{Q}_k^{\tilde{\pi}_k}(s_0,\cdot) \right\rangle - \left\langle \pi^\star(\cdot \mid s_0), \widehat{Q}_k^{\pi^\star}(s_0,\cdot) \right\rangle \\
&= \left\langle \tilde{\pi}_k(\cdot \mid s_0) - \pi^\star(\cdot \mid s_0), \widehat{Q}_k^{\tilde{\pi}_k}(s_0,\cdot) \right\rangle + \left\langle \pi^\star(\cdot \mid s_0), \widehat{Q}_k^{\tilde{\pi}_k}(s_0,\cdot) - \widehat{Q}_k^{\pi^\star}(s_0,\cdot) \right\rangle \\
&= \left\langle \tilde{\pi}_k(\cdot \mid s_0) - \pi^\star(\cdot \mid s_0), \widehat{Q}_k^{\tilde{\pi}_k}(s_0,\cdot) \right\rangle + \left\langle \pi^\star(\cdot \mid s_0), \gamma\left[ \widehat{P}_{k_i}\left( \widehat{V}_k^{\tilde{\pi}_k} - \widehat{V}_k^{\pi^\star} \right) \right](s_0,\cdot) \right\rangle \\
&= \mathbb{E}\left[ \sum_{\tau=0}^{\infty} \gamma^\tau \left\langle \tilde{\pi}_k(\cdot \mid s_{k,\tau}) - \pi^\star(\cdot \mid s_{k,\tau}), \widehat{Q}_k^{\tilde{\pi}_k}(s_{k,\tau},\cdot) \right\rangle \,\middle|\, \pi^\star, \widehat{P}_{k_i}, s_0 \right],
\end{aligned}
$$

where recall $\widehat{Q}_k^\pi(s,a) = \mathbb{E}\left[ \sum_{\tau=0}^{\infty} \gamma^t[\ell_k - \widehat{b}_k](s_{k,\tau},a_{k,\tau}) \,\middle|\, \pi, \widehat{P}_{k_i}, (s_{k,0},a_{k,0}) = (s,a) \right]$ is the state-action value of policy $\pi$ under the empirical model $(\widehat{P}_{k_i}, \ell_k - \widehat{b}_{k_i})$ and the expectation $\mathbb{E}[\cdot \mid \pi^\star, \widehat{P}_{k_i}, s_0]$ is taken over the randomness of the state-action sequence $\{(s_{k,\tau}, a_{k,\tau})\}_{\tau=0}^{\infty}$ with $a_{k,\tau} \sim \pi^\star(\cdot \mid s_{k,\tau})$, $s_{k,\tau+1} \sim \widehat{P}_{k_i}(\cdot \mid s_{k,\tau},a_{k,\tau})$, and $s_{k,0} = s_0$.

Taking summation of the above display over $k \in \{k_i, k_i + 1, \ldots, k_i + L - 1\}$ and re-arranging show that

$$
\begin{aligned}
&\mathbb{E}\left[ \sum_{k=k_i}^{k_i+L-1} \left( \widehat{V}_k^{\tilde{\pi}_k}(s_0) - \widehat{V}_k^{\pi^\star}(s_0) \right) \right] \\
&= \mathbb{E}\left[ \mathbb{E}\left[ \sum_{k=k_i}^{k_i+L-1} \sum_{\tau=0}^{\infty} \gamma^\tau \left\langle \tilde{\pi}_k(\cdot \mid s_{k,\tau}) - \pi^\star(\cdot \mid s_{k,\tau}), \widehat{Q}_k^{\tilde{\pi}_k}(s_{k,\tau},\cdot) \right\rangle \,\middle|\, \pi^\star, \widehat{P}_{k_i}, s_0 \right] \right] \\
&= \sum_{\tau=0}^{\infty} \gamma^\tau \mathbb{E}\left[ \mathbb{E}\left[ \sum_{k=k_i}^{k_i+L-1} \left\langle \tilde{\pi}_k(\cdot \mid s_{k,\tau}) - \pi^\star(\cdot \mid s_{k,\tau}), \widehat{Q}_k^{\tilde{\pi}_k}(s_{k,\tau},\cdot) \right\rangle \,\middle|\, \pi^\star, \widehat{P}_{k_i}, s_0 \right] \right]. \quad (6)
\end{aligned}
$$

Further, note that the update process of policy in Eq. (3) can be solved by the following two-step procedure [Lattimore and Szepesvári, 2020]:

$$\widehat{\pi}_{k+1}(\cdot \mid s) \in \underset{\pi(\cdot \mid s) \in \mathbb{R}_+^A}{\arg\min} \ \eta \left\langle \pi(\cdot \mid s), \widehat{Q}_k^{\tilde{\pi}_k}(s, \cdot) \right\rangle + D_F(\pi(\cdot \mid s), \tilde{\pi}_k(\cdot \mid s)) \quad \text{and} \tag{7}$$

$$\tilde{\pi}_{k+1}(\cdot \mid s) \in \underset{\pi(\cdot \mid s) \in \Delta(\mathcal{A})}{\arg\min} \ D_F(\pi(\cdot \mid s), \widehat{\pi}_{k+1}(\cdot \mid s)), \tag{8}$$

for any $s \in \mathcal{S}$. Eq. (7) combined with the first-order optimality condition implies that

$$\widehat{Q}_k^{\tilde{\pi}_k}(s, \cdot) = -\frac{1}{\eta}(\nabla F(\widehat{\pi}_{k+1}(\cdot \mid s)) - \nabla F(\tilde{\pi}_k(\cdot \mid s))), \tag{9}$$

for any $s \in \mathcal{S}$. This display shows that $\widehat{\pi}_{k+1}(a \mid s) = \tilde{\pi}_k(a \mid s) \exp(-\eta \widehat{Q}_k^{\tilde{\pi}_k}(s, \cdot))$, and $\widehat{\pi}_{k+1}(a \mid s) \le \tilde{\pi}_k(a \mid s)$ since $\widehat{Q}_k^{\tilde{\pi}_k}(s, a) \ge 0$, for any $(s, a) \in \mathcal{S} \times \mathcal{A}$.

Therefore, one can see that

$$\mathbb{E}\left[\left\langle \tilde{\pi}_k(\cdot \mid s_{k,\tau}) - \pi^\star(\cdot \mid s_{k,\tau}), \widehat{Q}_k^{\tilde{\pi}_k}(s_{k,\tau}, \cdot) \right\rangle \Big| \pi^\star, \widehat{P}_{k_i}, s_0\right]$$

$$= \mathbb{E}\left[\frac{1}{\eta} \langle \pi^\star(\cdot \mid s_{k,\tau}) - \tilde{\pi}_k(\cdot \mid s_{k,\tau}), \nabla F(\widehat{\pi}_{k+1}(\cdot \mid s_{k,\tau})) - \nabla F(\tilde{\pi}_k(\cdot \mid s_{k,\tau})) \rangle \Big| \pi^\star, \widehat{P}_{k_i}, s_0\right]$$

$$= \mathbb{E}\left[\frac{1}{\eta}(D_F(\pi^\star(\cdot \mid s_{k,\tau}), \tilde{\pi}_k(\cdot \mid s_{k,\tau})) + D_F(\tilde{\pi}_k(\cdot \mid s_{k,\tau}), \widehat{\pi}_{k+1}(\cdot \mid s_{k,\tau}))\right.$$
$$\left. - D_F(\pi^\star(\cdot \mid s_{k,\tau}), \widehat{\pi}_{k+1}(\cdot \mid s_{k,\tau}))) \Big| \pi^\star, \widehat{P}_{k_i}, s_0\right]$$

$$\le \mathbb{E}\left[\frac{1}{\eta}(D_F(\pi^\star(\cdot \mid s_{k,\tau}), \tilde{\pi}_k(\cdot \mid s_{k,\tau})) + D_F(\tilde{\pi}_k(\cdot \mid s_{k,\tau}), \widehat{\pi}_{k+1}(\cdot \mid s_{k,\tau}))\right.$$
$$\left. - D_F(\pi^\star(\cdot \mid s_{k,\tau}), \tilde{\pi}_{k+1}(\cdot \mid s_{k,\tau}))) \Big| \pi^\star, \widehat{P}_{k_i}, s_0\right], \tag{10}$$

where the first equality comes from Eq. (9), the second equality is due to the three-point lemma, and the last inequality follows the generalized Pythagorean theorem.

Taking summation of Eq. (10) over $k \in \{k_i, k_i + 1, \ldots, k_i + L - 1\}$ leads to

$$\mathbb{E}\left[\sum_{k=k_i}^{k_i+L-1} \left\langle \tilde{\pi}_k(\cdot \mid s_{k,\tau}) - \pi^\star(\cdot \mid s_{k,\tau}), \widehat{Q}_k^{\tilde{\pi}_k}(s_{k,\tau}, \cdot) \right\rangle \Big| \pi^\star, \widehat{P}_{k_i}, s_0\right]$$

$$\le \mathbb{E}\left[\frac{1}{\eta}\left(D_F(\pi^\star(\cdot \mid s_{k_i,\tau}), \tilde{\pi}_{k_i}(\cdot \mid s_{k_i,\tau})) + \sum_{k=k_i}^{k_i+L-1} D_F(\tilde{\pi}_k(\cdot \mid s_{k,\tau}), \widehat{\pi}_{k+1}(\cdot \mid s_{k,\tau}))\right) \Big| \pi^\star, \widehat{P}_{k_i}, s_0\right]. \tag{11}$$

The first term in Eq. (11) can be bounded as follows:

$$\mathbb{E}\left[D_F(\pi^\star(\cdot \mid s_{k_i,\tau}), \tilde{\pi}_{k_i}(\cdot \mid s_{k_i,\tau})) \Big| \pi^\star, \widehat{P}_{k_i}, s_0\right]$$

$$= \mathbb{E}\left[\sum_{a \in \mathcal{A}} \pi^\star(a \mid s_{k_i,\tau}) \ln \frac{\pi^\star(a \mid s_{k_i,\tau})}{\tilde{\pi}_{k_i}(a \mid s_{k_i,\tau})} \Big| \pi^\star, \widehat{P}_{k_i}, s_0\right]$$

$$\le \mathbb{E}\left[\sum_{a \in \mathcal{A}} \pi^\star(a \mid s_{k_i,\tau}) \ln A \Big| \pi^\star, \widehat{P}_{k_i}, s_0\right]$$

$$= \ln A, \tag{12}$$

where the inequality is because we choose $\tilde{\pi}_{k_i}(\cdot \mid s) = U(\mathcal{A})$ for any $s \in \mathcal{S}$ in Algorithm 1.

It remains to bound $\mathbb{E}\left[\sum_{k=k_i}^{k_i+L-1} D_F(\tilde{\pi}_k(\cdot \mid s_{k,\tau}), \widehat{\pi}_{k+1}(\cdot \mid s_{k,\tau})) \Big| \pi^\star, \widehat{P}_{k_i}, s_0\right]$ in Eq. (11):

$$\mathbb{E}\left[\sum_{k=k_i}^{k_i+L-1} D_F(\tilde{\pi}_k(\cdot \mid s_{k,\tau}), \widehat{\pi}_{k+1}(\cdot \mid s_{k,\tau})) \Big| \pi^\star, \widehat{P}_{k_i}, s_0\right]$$

$$
\begin{aligned}
=&\mathbb{E}\left[\sum_{k=k_i}^{k_i+L-1}\left(-D_F(\widehat{\pi}_{k+1}(\cdot\mid s_{k,\tau}),\tilde{\pi}_k(\cdot\mid s_{k,\tau}))\right.\right.\\
&\left.\left.+\langle\nabla F(\tilde{\pi}_k(\cdot\mid s_{k,\tau}))-\nabla F(\widehat{\pi}_{k+1}(\cdot\mid s_{k,\tau})),\tilde{\pi}_k(\cdot\mid s_{k,\tau})-\widehat{\pi}_{k+1}(\cdot\mid s_{k,\tau})\rangle\right)\right|\pi^\star,\widehat{P}_{k_i},s_0\right]\\
\leq&\mathbb{E}\left[\sum_{k=k_i}^{k_i+L-1}\left(-D_F(\widehat{\pi}_{k+1}(\cdot\mid s_{k,\tau}),\tilde{\pi}_k(\cdot\mid s_{k,\tau}))+\frac{1}{2}\|\nabla F(\tilde{\pi}_k(\cdot\mid s_{k,\tau}))-\nabla F(\widehat{\pi}_{k+1}(\cdot\mid s_{k,\tau}))\|^2_{\nabla^{-2}F(z_k(\cdot\mid s_{k,\tau}))}\right.\right.\\
&\left.\left.+\frac{1}{2}\|\tilde{\pi}_k(\cdot\mid s_{k,\tau})-\widehat{\pi}_{k+1}(\cdot\mid s_{k,\tau})\|^2_{\nabla^2 F(z_k(\cdot\mid s_{k,\tau}))}\right)\right|\pi^\star,\widehat{P}_{k_i},s_0\right]\\
=&\mathbb{E}\left[\sum_{k=k_i}^{k_i+L-1}\left(-D_F(\widehat{\pi}_{k+1}(\cdot\mid s_{k,\tau}),\tilde{\pi}_k(\cdot\mid s_{k,\tau}))+\frac{1}{2}\|\eta\widehat{Q}_k^{\tilde{\pi}_k}(s_{k,\tau},\cdot)\|^2_{\nabla^{-2}F(z_k(\cdot\mid s_{k,\tau}))}\right.\right.\\
&\left.\left.+\frac{1}{2}\|\tilde{\pi}_k(\cdot\mid s_{k,\tau})-\widehat{\pi}_{k+1}(\cdot\mid s_{k,\tau})\|^2_{\nabla^2 F(z_k(\cdot\mid s_{k,\tau}))}\right)\right|\pi^\star,\widehat{P}_{k_i},s_0\right]\\
=&\mathbb{E}\left[\sum_{k=k_i}^{k_i+L-1}\left(-\frac{1}{2}\|\tilde{\pi}_k(\cdot\mid s_{k,\tau})-\widehat{\pi}_{k+1}(\cdot\mid s_{k,\tau})\|^2_{\nabla^2 F(\omega_k(\cdot\mid s_{k,\tau}))}+\frac{1}{2}\|\eta\widehat{Q}_k^{\tilde{\pi}_k}(s_{k,\tau},\cdot)\|^2_{\nabla^{-2}F(z_k(\cdot\mid s_{k,\tau}))}\right.\right.\\
&\left.\left.+\frac{1}{2}\|\tilde{\pi}_k(\cdot\mid s_{k,\tau})-\widehat{\pi}_{k+1}(\cdot\mid s_{k,\tau})\|^2_{\nabla^2 F(z_k(\cdot\mid s_{k,\tau}))}\right)\right|\pi^\star,\widehat{P}_{k_i},s_0\right]\\
=&\mathbb{E}\left[\frac{\eta^2}{2}\sum_{k=k_i}^{k_i+L-1}\sum_{a\in\mathcal{A}}z_k(a\mid s_{k,\tau})\widehat{Q}_k^{\tilde{\pi}_k}(s_{k,\tau},a)^2\right|\pi^\star,\widehat{P}_{k_i},s_0\right]\\
\leq&\mathbb{E}\left[\frac{\eta^2}{2}\sum_{k=k_i}^{k_i+L-1}\sum_{a\in\mathcal{A}}\tilde{\pi}_k(a\mid s_{k,\tau})\widehat{Q}_k^{\tilde{\pi}_k}(s_{k,\tau},a)^2\right|\pi^\star,\widehat{P}_{k_i},s_0\right]\\
\leq&2\eta^2 L/(1-\gamma)^2,
\end{aligned}
\tag{13}
$$

where the second line comes from the Young-Fenchel inequality for all $z_k(\cdot\mid s_{k,\tau})\in[\widehat{\pi}_{k+1}(\cdot\mid s_{k,\tau}),\tilde{\pi}_k(\cdot\mid s_{k,\tau})]$ arbitrarily, the third line follows by the first-order optimality condition in Eq. (9), the fourth line is by the mean value theorem of the second derivative for some fixed $\omega_k(\cdot\mid s_{k,\tau})\in[\widehat{\pi}_{k+1}(\cdot\mid s_{k,\tau}),\tilde{\pi}_k(\cdot\mid s_{k,\tau})]$, the fifth line comes from fixing $z_k(\cdot\mid s_{k,\tau})=\omega_k(\cdot\mid s_{k,\tau})$, the sixth line comes from the fact that $z_k(\cdot\mid s_{k,\tau})\leq\tilde{\pi}_k(\cdot\mid s_{k,\tau})$ and the last line is due to that $|\widehat{Q}_k^{\tilde{\pi}_k}(s_{k,\tau},a)|\leq 2/(1-\gamma)$.

Substituting Eq. (12) and Eq. (13) into Eq. (11), along with Eq. (6), shows that

$$
\begin{aligned}
&\mathbb{E}\left[\sum_{k=1}^K\left(\widehat{V}_k^{\tilde{\pi}_k}-\widehat{V}_k^{\pi^\star}\right)\right]\\
=&\mathbb{E}\left[\mathbb{E}_{s_0\sim d_0}\left[\sum_{k=1}^K\left(\widehat{V}_k^{\tilde{\pi}_k}(s_0)-\widehat{V}_k^{\pi^\star}(s_0)\right)\right]\right]\\
=&\mathbb{E}\left[\mathbb{E}_{s_0\sim d_0}\left[\sum_{i=1}^N\sum_{k=k_i}^{k_i+L-1}\left(\widehat{V}_k^{\tilde{\pi}_k}(s_0)-\widehat{V}_k^{\pi^\star}(s_0)\right)\right]\right]\\
=&\sum_{\tau=0}^\infty\gamma^\tau\mathbb{E}\left[\mathbb{E}_{s_0\sim d_0}\left[\sum_{i=1}^N\mathbb{E}\left[\sum_{k=k_i}^{k_i+L-1}\left\langle\tilde{\pi}_k(\cdot\mid s_{k,\tau})-\pi^\star(\cdot\mid s_{k,\tau}),\widehat{Q}_k^{\tilde{\pi}_k}(s_{k,\tau},\cdot)\right\rangle\right|\pi^\star,\widehat{P}_{k_i},s_0\right]\right]\right]\\
\leq&\sum_{\tau=0}^\infty\gamma^\tau N\cdot\frac{1}{\eta}\left(\ln A+2\eta^2 L/(1-\gamma)^2\right)\leq\frac{K\sqrt{2\ln A}}{\sqrt{L}(1-\gamma)^2},
\end{aligned}
$$

which completes the proof. $\qquad\square$

## A.2 Bounding OPTIMISM TERM

We now turn to prove Lemma 4.2, which provides a (near) optimism at the initial state distribution.

*Proof of Lemma 4.2.* Recall $f_{k_i}(s,a) = \left\| \widehat{P}_{k_i}(\cdot \mid s,a) - P^\star(\cdot \mid s,a) \right\|_1$. In the following, we condition on the good event

$$\mathcal{E} = \Big\{ \forall i \in [N], \mathbb{E}_{s \sim \rho_{k_i}, a \sim \bar{\pi}_{k_i}} \left[ f_{k_i}^2(s,a) \right] \leq \zeta_{k_i}, \mathbb{E}_{s \sim \rho'_{k_i}, a \sim \bar{\pi}_{k_i}} \left[ f_{k_i}^2(s,a) \right] \leq \zeta_{k_i} ;$$

$$\forall i \in [N], \forall \phi, \|\phi(s,a)\|_{\widehat{\Sigma}^{-1}_{k_i,\phi}} = \Theta \left( \|\phi(s,a)\|_{\Sigma^{-1}_{\rho_{k_i} \times \bar{\pi}_{k_i}, \phi}} \right) \Big\},$$

which is guaranteed to hold with probability $1 - \delta$ by using union bound with Lemma C.1 and Lemma C.3.

We first consider fixed initial state $s_0$. For some epoch $i \in [N]$ and episode $k \in \{k_i, k_i+1, \ldots, k_i + L - 1\}$, applying Eq. (32) in Lemma C.4 implies that

$$\widehat{V}_k^{\pi^\star}(s_0) - V_k^{\pi^\star}(s_0)$$

$$= \frac{1}{1-\gamma} \mathbb{E}_{(s,a) \sim d_{\widehat{P}_{k_i}}^{\pi^\star}} \left[ -\widehat{b}_{k_i}(s,a) + \gamma \left( \widehat{P}_{k_i}(\cdot \mid s,a) - P^\star(\cdot \mid s,a) \right)^\top V_k^{\pi^\star} \right]$$

$$\leq \frac{1}{1-\gamma} \mathbb{E}_{(s,a) \sim d_{\widehat{P}_{k_i}}^{\pi^\star}} \left[ -\widehat{b}_{k_i}(s,a) + \frac{\gamma}{1-\gamma} f_{k_i}(s,a) \right]$$

$$= \frac{1}{1-\gamma} \mathbb{E}_{(s,a) \sim d_{\widehat{P}_{k_i}}^{\pi^\star}} \left[ -\min \left( \alpha_{k_i} \left\| \widehat{\phi}_{k_i}(s,a) \right\|_{\Sigma^{-1}_{\rho_{k_i} \times \bar{\pi}_{k_i}, \widehat{\phi}_{k_i}}}, 2 \right) \frac{1}{(1-\gamma)} \right]$$

$$+ \frac{\gamma}{(1-\gamma)^2} \Bigg( \underbrace{\gamma \mathbb{E}_{(\tilde{s},\tilde{a}) \sim d_{\widehat{P}_{k_i}}^{\pi^\star}, s \sim \widehat{P}_{k_i}(\cdot \mid \tilde{s},\tilde{a}), a \sim \pi^\star(s)} \left[ f_{k_i}(s,a) \right]}_{\text{TERM}_1} + \underbrace{(1-\gamma) \mathbb{E}_{s \sim d_0, a \sim \pi^\star(s)} \left[ f_{k_i}(s,a) \right]}_{\text{TERM}_2} \Bigg),$$

$$\tag{14}$$

where the inequality comes from the Cauchy–Schwarz inequality together with $\|V_k^{\pi^\star}\|_\infty = 1/(1-\gamma)$, and the second equality is due to the fact that $\widehat{\Sigma}_{k_i, \widehat{\phi}_{k_i}}$ is an unbiased estimate of $\Sigma_{\rho_{k_i} \times \bar{\pi}_{k_i}, \widehat{\phi}_{k_i}}$ and Lemma A.4.

We first bound $\text{TERM}_2$ as follows:

$$(1-\gamma) \mathbb{E}_{s \sim d_0, a \sim \pi^\star(s)} \left[ f_{k_i}(s,a) \right] \leq \sqrt{ \frac{(1-\gamma)A}{\xi} \mathbb{E}_{s \sim \rho_{k_i}, a \sim \bar{\pi}_{k_i}} \left[ f_{k_i}^2(s,a) \right]} \leq \sqrt{ \frac{(1-\gamma)A\zeta_{k_i}}{\xi}},$$

$$\tag{15}$$

where the first inequality is due to Lemma A.1 and the second inequality is by the definition of event $\mathcal{E}$.

It remains to bound $\text{TERM}_1$:

$$\gamma \mathbb{E}_{(\tilde{s},\tilde{a}) \sim d_{\widehat{P}_{k_i}}^{\pi^\star}, s \sim \widehat{P}_{k_i}(\cdot \mid \tilde{s},\tilde{a}), a \sim \pi^\star(s)} \left[ f_{k_i}(s,a) \right]$$

$$\leq \gamma \mathbb{E}_{(\tilde{s},\tilde{a}) \sim d_{\widehat{P}_{k_i}}^{\pi^\star}} \left[ \left\| \widehat{\phi}_{k_i}(\tilde{s},\tilde{a}) \right\|_{\Sigma^{-1}_{\rho_{k_i} \times \bar{\pi}_{k_i}, \widehat{\phi}_{k_i}}} \right] \sqrt{ \frac{k_i A}{\xi} \mathbb{E}_{s \sim \rho'_{k_i}, a \sim \bar{\pi}_{k_i}} \left[ f_{k_i}^2(s,a) \right] + 4\lambda_{k_i} d + 4k_i \zeta_{k_i}}$$

$$\leq \gamma \mathbb{E}_{(\tilde{s},\tilde{a}) \sim d_{\widehat{P}_{k_i}}^{\pi^\star}} \left[ \left\| \widehat{\phi}_{k_i}(\tilde{s},\tilde{a}) \right\|_{\Sigma^{-1}_{\rho_{k_i} \times \bar{\pi}_{k_i}, \widehat{\phi}_{k_i}}} \right] \sqrt{ \frac{k_i A}{\xi} \zeta_{k_i} + 4\lambda_{k_i} d + 4k_i \zeta_{k_i}}$$

$$\lesssim \alpha_{k_i} \mathbb{E}_{(\tilde{s},\tilde{a}) \sim d_{\widehat{P}_{k_i}}^{\pi^\star}} \left[ \left\| \widehat{\phi}_{k_i}(\tilde{s},\tilde{a}) \right\|_{\Sigma^{-1}_{\rho_{k_i} \times \bar{\pi}_{k_i}, \widehat{\phi}_{k_i}}} \right],$$

$$\tag{16}$$

where the first inequality follows by Lemma A.3 as well as $\|f_{k_i}\|_\infty \leq 2$, the second inequality is again due to the definition of the good event $\mathcal{E}$, and the last inequality comes from the definition of $\alpha_{k_i}$.

Now substituting Eq. (15) and Eq. (16) into Eq. (14) shows that

$$\sum_{k=1}^{K}\left(\widehat{V}_k^{\pi^\star} - V_k^{\pi^\star}\right)$$

$$=\mathbb{E}_{s_0 \sim d_0}\left[\sum_{k=1}^{K}\left(\widehat{V}_k^{\pi^\star}(s_0) - V_k^{\pi^\star}(s_0)\right)\right]$$

$$=\mathbb{E}_{s_0 \sim d_0}\left[\sum_{i=1}^{N}\sum_{k=k_i}^{k_i+L-1}\left(\widehat{V}_k^{\pi^\star}(s_0) - V_k^{\pi^\star}(s_0)\right)\right]$$

$$\leq \sum_{i=1}^{N}\sum_{k=k_i}^{k_i+L-1}\frac{1}{(1-\gamma)^2}\mathbb{E}_{(s,a)\sim d_{\widehat{P}_{k_i}}^{\pi^\star}}\left[-\min\left(\alpha_{k_i}\left\|\widehat{\phi}_{k_i}(s,a)\right\|_{\Sigma_{\rho_{k_i}\times\bar{\pi}_{k_i},\widehat{\phi}_{k_i}}^{-1}},2\right)\right.$$

$$\left.+\min\left(\alpha_{k_i}\left\|\widehat{\phi}_{k_i}(s,a)\right\|_{\Sigma_{\rho_{k_i}\times\bar{\pi}_{k_i},\widehat{\phi}_{k_i}}^{-1}}+\sqrt{\frac{(1-\gamma)A\zeta_{k_i}}{\xi}},2\right)\right]$$

$$\leq \sum_{i=1}^{N}L\sqrt{\frac{A\ln(MN/\delta)}{\xi(1-\gamma)^3 k_i}}$$

$$=\sum_{i=1}^{N}L\sqrt{\frac{A\ln(MN/\delta)}{\xi(1-\gamma)^3((i-1)L+1)}}$$

$$\leq(L+\sqrt{K})\sqrt{\frac{A\ln(MN/\delta)}{\xi(1-\gamma)^3}},$$

where the first inequality follows by $\|f_{k_i}\|_\infty \leq 2$ for any $i \in [N]$ and the second inequality is due to the definition of $\zeta_{k_i}$ in Lemma C.1. $\qquad\square$

## A.3 Bounding ESTIMATION BIAS TERM

We now give the proof of Lemma 4.3, which controls the estimation bias term.

*Proof of Lemma 4.3.* Similar to the proof of Lemma 4.2, in what follows, we condition on the good event

$$\mathcal{E} = \left\{\forall i \in [N], \mathbb{E}_{s\sim\rho_{k_i},a\sim\bar{\pi}_{k_i}}\left[f_{k_i}^2(s,a)\right] \leq \zeta_{k_i}, \mathbb{E}_{s\sim\rho'_{k_i},a\sim\bar{\pi}_{k_i}}\left[f_{k_i}^2(s,a)\right] \leq \zeta_{k_i};\right.$$

$$\left.\forall i \in [N], \forall\phi, \|\phi(s,a)\|_{\widehat{\Sigma}_{k_i,\phi}^{-1}} = \Theta\left(\|\phi(s,a)\|_{\Sigma_{\rho_{k_i}\times\bar{\pi}_{k_i},\phi}^{-1}}\right)\right\}.$$

We first consider fixed initial state $s_0$. For some epoch $i \in [N]$ and episode $k \in \{k_i, k_i + 1, \ldots, k_i + L - 1\}$, applying Lemma C.4 shows that

$$V_k^{\bar{\pi}_k}(s_0) - \widehat{V}_k^{\bar{\pi}_k}(s_0) \tag{17}$$

$$=(1-\gamma)^{-1}\mathbb{E}_{(s,a)\sim d_{P^\star}^{\bar{\pi}_k}}\left[\widehat{b}_{k_i}(s,a) - \gamma\mathbb{E}_{\widehat{P}_{k_i}(s'|s,a)}\left[\widehat{V}_k^{\bar{\pi}_k}(s')\right] + \gamma\mathbb{E}_{P^\star(s'|s,a)}\left[\widehat{V}_k^{\bar{\pi}_k}(s')\right]\right]$$

$$\leq\mathbb{E}_{(s,a)\sim d_{P^\star}^{\bar{\pi}_k}}\left[\frac{1}{1-\gamma}\widehat{b}_{k_i}(s,a) + \frac{2}{(1-\gamma)^3}f_{k_i}(s,a)\right]$$

$$=\underbrace{\gamma\mathbb{E}_{(\tilde{s},\tilde{a})\sim d_{P^\star}^{\bar{\pi}_k},s\sim P^\star(\cdot|\tilde{s},\tilde{a}),a\sim\bar{\pi}_k(s)}\left[\frac{1}{1-\gamma}\widehat{b}_{k_i}(s,a) + \frac{2}{(1-\gamma)^3}f_{k_i}(s,a)\right]}_{\text{TERM}_1}.$$

$$+ (1-\gamma)\mathbb{E}_{s\sim d_0, a\sim\tilde{\pi}_k(s_0)}\underbrace{\left[\frac{1}{1-\gamma}\widehat{b}_{k_i}(s,a) + \frac{2}{(1-\gamma)^3}f_{k_i}(s,a)\right]}_{\text{TERM}_2}, \tag{18}$$

where the inequality follows from the fact that $\|\widehat{b}_{k_i}\|_\infty \le 2/(1-\gamma)$ and $\|\widehat{V}_{k_i}^{\tilde{\pi}_k}\|_\infty \le 2/(1-\gamma)^2$ and the second equality is due to Lemma A.4.

$\text{TERM}_2$ can be bounded as follows:

$$\begin{aligned}
\text{TERM}_2 &= \mathbb{E}_{s\sim d_0, a\sim\tilde{\pi}_k(s_0)}\left[\widehat{b}_{k_i}(s,a) + \frac{2}{(1-\gamma)^2}f_{k_i}(s,a)\right]\\
&\le \sqrt{\frac{A}{(1-\gamma)\xi}\mathbb{E}_{s\sim\rho_{k_i}, a\sim\bar{\pi}_{k_i}(s)}\left[\widehat{b}_{k_i}^2(s,a) + \frac{4}{(1-\gamma)^4}f_{k_i}^2(s,a)\right]}\\
&\lesssim \sqrt{\frac{dA\alpha_{k_i}^2}{(1-\gamma)^3 k_i\xi} + \frac{A\zeta_{k_i}}{(1-\gamma)^5\xi}}\\
&\le \sqrt{\frac{dA\alpha_{k_i}^2}{(1-\gamma)^3 k_i\xi}} + \sqrt{\frac{A\zeta_{k_i}}{(1-\gamma)^5\xi}}, \tag{19}
\end{aligned}$$

where the first inequality follows from Lemma A.1, and the second inequality comes from Lemma C.1 as well as the following inequality:

$$\begin{aligned}
&\mathbb{E}_{s\sim\rho_{k_i}, a\sim\bar{\pi}_{k_i}(s)}\left[k_i\widehat{b}_{k_i}^2(s,a)\right]\\
\le& \frac{k_i\alpha_{k_i}^2}{(1-\gamma)^2}\mathbb{E}_{s\sim\rho_{k_i}, a\sim\bar{\pi}_{k_i}(s)}\left[\left\|\widehat{\phi}_{k_i}(s,a)\right\|_{\Sigma_{\rho_{k_i}\times\bar{\pi}_{k_i},\widehat{\phi}_{k_i}}^{-1}}^2\right]\\
=& \frac{k_i\alpha_{k_i}^2}{(1-\gamma)^2}\text{Tr}\left(\mathbb{E}_{s\sim\rho_{k_i}, a\sim\bar{\pi}_{k_i}}\left[\widehat{\phi}_{k_i}(s,a)\widehat{\phi}_{k_i}(s,a)^\top\right]\left\{k_i\mathbb{E}_{s\sim\rho_{k_i}, a\sim\bar{\pi}_{k_i}}\left[\widehat{\phi}_{k_i}(s,a)\widehat{\phi}_{k_i}(s,a)^\top\right] + \lambda_{k_i}I\right\}^{-1}\right)\\
\le& \frac{\alpha_{k_i}^2 d}{(1-\gamma)^2}, \tag{20}
\end{aligned}$$

where the first inequality is because $\widehat{\Sigma}_{k_i,\phi_{k_i}}$ is an unbiased estimate of $\Sigma_{\rho_{k_i}\times\bar{\pi}_{k_i},\phi_{k_i}}$ and the second inequality is by $\text{Tr}(AB) \ge \text{Tr}(AC)$ for positive semi-definite matrices $A$, $B$, $C$ and $B - C \succeq 0$.

To bound $\text{TERM}_1$, we note that

$$\begin{aligned}
&\mathbb{E}_{(\tilde{s},\tilde{a})\sim d_{P^\star}^{\tilde{\pi}_k}, s\sim P^\star(\cdot|\tilde{s},\tilde{a}), a\sim\tilde{\pi}_k(s)}\left[\widehat{b}_{k_i}(s,a) + \frac{2}{(1-\gamma)^2}f_{k_i}(s,a)\right]\\
\le& \mathbb{E}_{(\tilde{s},\tilde{a})\sim d_{P^\star}^{\tilde{\pi}_k}}\left[\|\phi^\star(\tilde{s},\tilde{a})\|_{\Sigma_{\rho_{k_i},\phi^\star}^{-1}}\right]\sqrt{\frac{2k_iA}{\xi\gamma}\mathbb{E}_{s\sim\rho_{k_i}, a\sim\bar{\pi}_{k_i}(s)}\left[\widehat{b}_{k_i}^2(s,a) + \frac{4}{(1-\gamma)^4}f_{k_i}^2(s,a)\right] + \lambda_{k_i}d\frac{36}{(1-\gamma)^4}}\\
\le& \mathbb{E}_{(\tilde{s},\tilde{a})\sim d_{P^\star}^{\tilde{\pi}_k}}\left[\|\phi^\star(\tilde{s},\tilde{a})\|_{\Sigma_{\rho_{k_i},\phi^\star}^{-1}}\right]\sqrt{\frac{2k_iA}{\xi\gamma}\left(\frac{\alpha_{k_i}^2 d}{k_i(1-\gamma)^2} + \frac{4}{(1-\gamma)^4}\zeta_{k_i}\right) + \lambda_{k_i}d\frac{36}{(1-\gamma)^4}}\\
\lesssim& \mathbb{E}_{(\tilde{s},\tilde{a})\sim d_{P^\star}^{\tilde{\pi}_k}}\left[\|\phi^\star(\tilde{s},\tilde{a})\|_{\Sigma_{\rho_{k_i},\phi^\star}^{-1}}\right]\sqrt{\frac{dA\alpha_{k_i}^2}{\xi\gamma(1-\gamma)^2}} + \mathbb{E}_{(\tilde{s},\tilde{a})\sim d_{P^\star}^{\tilde{\pi}_k}}\left[\|\phi^\star(\tilde{s},\tilde{a})\|_{\Sigma_{\rho_{k_i},\phi^\star}^{-1}}\right]\sqrt{\frac{k_iA\zeta_{k_i}}{\xi\gamma(1-\gamma)^4} + \frac{\lambda_{k_i}d}{\gamma(1-\gamma)^4}},
\end{aligned}$$

where the first inequality follows from Lemma A.2, the AM-GM inequality, $\|\widehat{b}_{k_i}\|_\infty \le 2(1-\gamma)$, and $\|f_{k_i}\|_\infty \le 2$ and the second inequality is due to Eq. (20).

The above display implies that $\text{TERM}_1$ can be bounded as follows:

$$\begin{aligned}
&\gamma\mathbb{E}_{(\tilde{s},\tilde{a})\sim d_{P^\star}^{\tilde{\pi}_k}, s\sim P^\star(\cdot|\tilde{s},\tilde{a}), a\sim\tilde{\pi}_k(s)}\left[\frac{1}{1-\gamma}\widehat{b}_{k_i}(s,a) + \frac{2}{(1-\gamma)^3}f_{k_i}(s,a)\right]\\
\lesssim& \frac{1}{(1-\gamma)^2}\mathbb{E}_{(\tilde{s},\tilde{a})\sim d_{P^\star}^{\tilde{\pi}_k}}\left[\|\phi^\star(\tilde{s},\tilde{a})\|_{\Sigma_{\rho_{k_i},\phi^\star}^{-1}}\right]\sqrt{\gamma\frac{dA\alpha_{k_i}^2}{\xi}} + \frac{1}{(1-\gamma)^3}\mathbb{E}_{(\tilde{s},\tilde{a})\sim d_{P^\star}^{\tilde{\pi}_k}}\left[\|\phi^\star(\tilde{s},\tilde{a})\|_{\Sigma_{\rho_{k_i},\phi^\star}^{-1}}\right]
\end{aligned}$$

$$\cdot \sqrt{\gamma\left(\frac{k_i A \zeta_{k_i}}{\xi} + \lambda_{k_i} d\right)}$$

$$\leq \frac{1}{(1-\gamma)^2} \mathbb{E}_{(\tilde{s},\tilde{a}) \sim d_{P^\star}^{\tilde{\pi}_k}}\left[\|\phi^\star(\tilde{s},\tilde{a})\|_{\Sigma_{\rho_{k_i},\phi^\star}^{-1}}\right]\sqrt{\frac{dA\alpha_K^2}{\xi}} + \frac{1}{(1-\gamma)^3}\mathbb{E}_{(\tilde{s},\tilde{a}) \sim d_{P^\star}^{\tilde{\pi}_k}}\left[\|\phi^\star(\tilde{s},\tilde{a})\|_{\Sigma_{\rho_{k_i},\phi^\star}^{-1}}\alpha_K\right]$$

$$\lesssim \frac{1}{(1-\gamma)^3}\mathbb{E}_{(\tilde{s},\tilde{a}) \sim d_{P^\star}^{\tilde{\pi}_k}}\left[\|\phi^\star(\tilde{s},\tilde{a})\|_{\Sigma_{\rho_{k_i},\phi^\star}^{-1}}\right]\sqrt{\frac{dA\alpha_K^2}{\xi}}, \tag{21}$$

where the second inequality is due to that $\alpha_k = O(\sqrt{\gamma(A/\xi + d^2)\ln(Mk/\delta)}) = O(\sqrt{\gamma(A/\xi + d^2)k\zeta_k}) = O(\sqrt{\gamma(Ak\zeta_k/\xi + \lambda_k d)}) = O(\sqrt{\gamma(Ak\zeta_k/\xi + \lambda_k d + k\zeta_k)})$.

Substituting Eq. (19) and Eq. (21) into Eq. (17), and taking summation over $k \in [K]$ leads to

$$\sum_{k=1}^K \left(V_k^{\tilde{\pi}_k} - \widehat{V}_k^{\tilde{\pi}_k}\right)$$

$$= \mathbb{E}_{s_0 \sim d_0}\left[\sum_{k=1}^K \left(V_k^{\tilde{\pi}_k}(s_0) - \widehat{V}_k^{\tilde{\pi}_k}(s_0)\right)\right]$$

$$\lesssim \sum_{i=1}^N \sum_{k=k_i}^{k_i+L-1}\left(\frac{1}{(1-\gamma)^3}\mathbb{E}_{(s,a)\sim d_{P^\star}^{\tilde{\pi}_k}}\left[\|\phi^\star(s,a)\|_{\Sigma_{\rho_{k_i},\phi^\star}^{-1}}\right]\sqrt{\frac{dA\alpha_K^2}{\xi}} + \sqrt{\frac{dA\alpha_K^2}{(1-\gamma)^3 k_i \xi}} + \sqrt{\frac{A\zeta_{k_i}}{(1-\gamma)^5 \xi}}\right)$$

$$\lesssim \frac{\alpha_K}{(1-\gamma)^3}\sqrt{\frac{dA}{\xi}}\cdot\sum_{i=1}^N \sum_{k=k_i}^{k_i+L-1}\mathbb{E}_{(s,a)\sim d_{P^\star}^{\tilde{\pi}_k}}\left[\|\phi^\star(s,a)\|_{\Sigma_{\rho_{k_i},\phi^\star}^{-1}}\right]$$

$$\leq \frac{\alpha_K}{(1-\gamma)^3}\sqrt{\frac{dA}{\xi}}\cdot\sqrt{K\sum_{i=1}^N \sum_{k=k_i}^{k_i+L-1}\mathbb{E}_{(s,a)\sim d_{P^\star}^{\tilde{\pi}_k}}\left[\|\phi^\star(s,a)\|_{\Sigma_{\rho_{k_i},\phi^\star}^{-1}}^2\right]}$$

$$= \frac{\alpha_K}{(1-\gamma)^3}\sqrt{\frac{dA}{\xi}}\cdot\sqrt{K\sum_{i=1}^N \sum_{k=k_i}^{k_i+L-1}\mathrm{Tr}\left(\mathbb{E}_{(s,a)\sim d_{P^\star}^{\tilde{\pi}_k}}\left[\phi^\star(s,a)\phi^\star(s,a)^\top\right]\Sigma_{\rho_{k_i},\phi^\star}^{-1}\right)}$$

$$= \frac{\alpha_K}{(1-\gamma)^3}\sqrt{\frac{dA}{\xi}}\cdot\sqrt{K\sum_{j=1}^L \sum_{i=1}^N \mathrm{Tr}\left(\mathbb{E}_{(s,a)\sim d_{P^\star}^{\tilde{\pi}_{(i-1)L+j}}}\left[\phi^\star(s,a)\phi^\star(s,a)^\top\right]\Sigma_{\rho_{k_i},\phi^\star}^{-1}\right)}$$

$$\leq \frac{\alpha_K}{(1-\gamma)^3}\sqrt{\frac{dA}{\xi}}$$

$$\cdot\sqrt{K\sum_{j=1}^L \sum_{i=1}^N \mathrm{Tr}\left(\mathbb{E}_{(s,a)\sim d_{P^\star}^{\tilde{\pi}_{(i-1)L+j}}}\left[\phi^\star(s,a)\phi^\star(s,a)^\top\right]\left(\sum_{q=1}^{i-1}\mathbb{E}_{(s,a)\sim d_{P^\star}^{\tilde{\pi}_{(q-1)L+j}}}\left[\phi^\star(s,a)\phi^\star(s,a)^\top\right] + \lambda_1 I\right)^{-1}\right)}$$

$$\lesssim \frac{\alpha_K}{(1-\gamma)^3}\sqrt{\frac{dA}{\xi}}\cdot\sqrt{K\sum_{j=1}^L d\ln\left(1 + \frac{N}{d\lambda_1}\right)}$$

$$\lesssim \frac{\alpha_K}{(1-\gamma)^3}\sqrt{\frac{dA}{\xi}}\cdot\sqrt{dLK\ln\left(1 + \frac{K}{d\lambda_1}\right)}$$

$$\lesssim \frac{d^2 A\sqrt{KL}}{\xi(1-\gamma)^3}\sqrt{\ln(1+K)\ln(MK/\delta)},$$

where the third inequality follows from Cauchy–Schwarz inequality together with Jensen's inequality, the fourth inequality is by $\mathrm{Tr}(AB) \geq \mathrm{Tr}(AC)$ for positive semi-definite matrices $A$, $B$, $C$ and $B - C \succeq 0$, and the fifth inequality is due to Lemma C.2. The proof is now completed. □

## A.4 One-step-back Inequalities

We first present the following lemma, which bounds the quantity under the initial state distribution, for any policy $\pi$. Note that this lemma holds for any $k \in [K]$.

**Lemma A.1.** *For any $g : \mathcal{S} \times \mathcal{A} \to \mathbb{R}$ such that $\|g\|_\infty \leq B$ and any policy $\pi$, it holds that*

$$\mathbb{E}_{s \sim d_0, a \sim \pi(s_0)} [g(s,a)] \leq \sqrt{\frac{A}{(1-\gamma)\xi} \mathbb{E}_{s \sim \rho_k, a \sim \bar{\pi}_k} [g^2(s,a)]}.$$

*Proof.*

$$\mathbb{E}_{s \sim d_0, a \sim \pi(s_0)} [g(s,a)] \leq \sqrt{\mathbb{E}_{s \sim d_0, a \sim \pi(s_0)} [g^2(s,a)]}$$

$$\leq \sqrt{\max_{(s,a) \in \mathcal{S} \times \mathcal{A}} \frac{d_0(s)\pi(a|s)}{\rho_k(s)\bar{\pi}_k(a|s)} \mathbb{E}_{s \sim \rho_k, a \sim \bar{\pi}_k} [g^2(s,a)]}$$

$$\leq \sqrt{\max_{(s,a) \in \mathcal{S} \times \mathcal{A}} \frac{d_0(s)\pi(a|s)}{(1-\gamma)d_0(s)\bar{\pi}_k(a|s)} \mathbb{E}_{s \sim \rho_k, a \sim \bar{\pi}_k} [g^2(s,a)]}$$

$$\leq \sqrt{\max_{(s,a) \in \mathcal{S} \times \mathcal{A}} \frac{1}{(1-\gamma)\xi \cdot U(a)} \mathbb{E}_{s \sim \rho_k, a \sim \bar{\pi}_k} [g^2(s,a)]}$$

$$\leq \sqrt{\frac{A}{(1-\gamma)\xi} \mathbb{E}_{s \sim \rho_k, a \sim \bar{\pi}_k} [g^2(s,a)]},$$

where the first inequality follows from Jensen's inequality, the second inequality is by importance sampling, and the fourth inequality is due to the definition of $\bar{\pi}_k$. $\square$

The following lemma shows that

$$\mathbb{E}_{(s,a) \sim d_{P^\star}^{\tilde{\pi}_k}} [g(s,a)] \lesssim \mathbb{E}_{(s,a) \sim d_{P^\star}^{\tilde{\pi}_k}} \left[ \|\phi^\star(s,a)\|_{\Sigma_{\rho_k,\phi^\star}^{-1}} \right],$$

if $\mathbb{E}_{s \sim \rho_k, a \sim \bar{\pi}_k} [g^2(s,a)]$ is upper bounded.

**Lemma A.2** (One-step-back inequality in the true model). *For any $g : \mathcal{S} \times \mathcal{A} \to \mathbb{R}$ such that $\|g\|_\infty \leq B$, any epoch $i \in [N]$ and any episode $k \in \{k_i, \ldots, k_i + L - 1\}$, it holds that*

$$\mathbb{E}_{(\tilde{s},\tilde{a}) \sim d_{P^\star}^{\tilde{\pi}_k}, s \sim P^\star(\cdot|\tilde{s},\tilde{a}), a \sim \tilde{\pi}_k(s)} [g(s,a)]$$

$$\leq \mathbb{E}_{(\tilde{s},\tilde{a}) \sim d_{P^\star}^{\tilde{\pi}_k}} \left[ \|\phi^\star(\tilde{s},\tilde{a})\|_{\Sigma_{\rho_{k_i},\phi^\star}^{-1}} \right] \sqrt{\frac{k_i A}{\xi\gamma} \mathbb{E}_{s \sim \rho_{k_i}, a \sim \bar{\pi}_{k_i}} [g^2(s,a)] + \lambda_{k_i} dB^2}.$$

*Proof.* To begin with, applying the Cauchy–Schwarz inequality shows that

$$\mathbb{E}_{(\tilde{s},\tilde{a}) \sim d_{P^\star}^{\tilde{\pi}_k}, s \sim P^\star(\cdot|\tilde{s},\tilde{a}), a \sim \tilde{\pi}_k(s)} [g(s,a)]$$

$$= \mathbb{E}_{(\tilde{s},\tilde{a}) \sim d_{P^\star}^{\tilde{\pi}_k}} \left[ \phi^\star(\tilde{s},\tilde{a})^\top \int \sum_a \mu^\star(s)\tilde{\pi}_k(a \mid s)g(s,a)d(s) \right]$$

$$\leq \mathbb{E}_{(\tilde{s},\tilde{a}) \sim d_{P^\star}^{\tilde{\pi}_k}} \left[ \|\phi^\star(\tilde{s},\tilde{a})\|_{\Sigma_{\rho_{k_i},\phi^\star}^{-1}} \left\| \int \sum_a \mu^\star(s)\tilde{\pi}_k(a \mid s)g(s,a)d(s) \right\|_{\Sigma_{\rho_{k_i},\phi^\star}} \right]. \quad (22)$$

We bound the second quadratic form w.r.t. $\Sigma_{\rho_{k_i},\phi^\star}$ in Eq. (22) as follows:

$$\left\| \int \sum_a \mu^\star(s)\tilde{\pi}_k(a \mid s)g(s,a)d(s) \right\|_{\Sigma_{\rho_{k_i},\phi^\star}}^2$$

$$= \left[\int \sum_a \mu^\star(s)\tilde{\pi}_k(a \mid s)g(s,a)d(s)\right]^\top \left\{k_i \mathbb{E}_{(s,a)\sim\rho_{k_i}}\left[\phi^\star(s,a)\phi^\star(s,a)^\top\right] + \lambda_k I\right\}$$
$$\left[\int \sum_a \mu^\star(s)\tilde{\pi}_k(a \mid s)g(s,a)d(s)\right]$$

$$\leq k_i \mathbb{E}_{(\tilde{s},\tilde{a})\sim\rho_{k_i}}\left\{\left[\int \sum_a \mu^\star(s)^\top \phi^\star(\tilde{s},\tilde{a})\tilde{\pi}_k(a \mid s)g(s,a)d(s)\right]^2\right\} + \lambda_{k_i}dB^2$$

$$= k_i \mathbb{E}_{(\tilde{s},\tilde{a})\sim\rho_{k_i}}\left\{\mathbb{E}_{s\sim P^\star(\cdot|\tilde{s},\tilde{a}),a\sim\tilde{\pi}_k(s)}\left[g(s,a)\right]^2\right\} + \lambda_{k_i}dB^2$$

$$\leq k_i \mathbb{E}_{(\tilde{s},\tilde{a})\sim\rho_{k_i},s\sim P^\star(\cdot|\tilde{s},\tilde{a}),a\sim\tilde{\pi}_k(s)}\left[g^2(s,a)\right] + \lambda_{k_i}dB^2 , \tag{23}$$

where the first inequality comes from $\|g(s,a)\|_\infty \leq B$ together with the regularity condition in Assumption 2.1 that $\left\|\int \mu^\star(s)h(s)\mathrm{d}(s)\right\|_2 \leq \sqrt{d}$ for any $h : \mathcal{S} \to [0,1]$, and the last inequality follows from the Jensen's inequality.

By importance sampling, it is clear that

$$k_i \mathbb{E}_{(\tilde{s},\tilde{a})\sim\rho_{k_i},s\sim P^\star(\cdot|\tilde{s},\tilde{a}),a\sim\tilde{\pi}_k(s)}\left[g^2(s,a)\right] + \lambda_{k_i}dB^2$$

$$\leq \frac{k_i}{\gamma}\mathbb{E}_{s\sim\rho_{k_i},a\sim\tilde{\pi}_k(s)}\left[g^2(s,a)\right] + \lambda_{k_i}dB^2$$

$$\leq \max_{(s,a)\in\mathcal{S}\times\mathcal{A}} \frac{k_i}{\gamma}\frac{\tilde{\pi}_k(a \mid s)}{\bar{\pi}_{k_i}(a \mid s)}\mathbb{E}_{s\sim\rho_{k_i},a\sim\bar{\pi}_{k_i}(s)}\left[g^2(s,a)\right] + \lambda_{k_i}dB^2$$

$$\leq \frac{k_i A}{\xi\gamma}\mathbb{E}_{s\sim\rho_{k_i},a\sim\bar{\pi}_{k_i}(s)}\left[g^2(s,a)\right] + \lambda_{k_i}dB^2 , \tag{24}$$

where the third inequality is due to the definition of $\bar{\pi}_k$, and the first inequality is because

$$\gamma\mathbb{E}_{(\tilde{s},\tilde{a})\sim\rho_{k_i},s\sim P^\star(\cdot|\tilde{s},\tilde{a}),a\sim\tilde{\pi}_k(s)}\left[g^2(s,a)\right]$$

$$\leq \gamma\mathbb{E}_{(\tilde{s},\tilde{a})\sim\rho_{k_i},s\sim P^\star(\cdot|\tilde{s},\tilde{a}),a\sim\tilde{\pi}_k(s)}\left[g^2(s,a)\right] + (1-\gamma)\mathbb{E}_{s_0\sim d_0,a\sim\tilde{\pi}_k(s)}\left[g^2(s,a)\right]$$

$$= \mathbb{E}_{s\sim\rho_{k_i},a\sim\tilde{\pi}_k(s)}\left[g^2(s,a)\right] ,$$

which comes from Lemma A.4.

The proof is concluded by substituting Eq. (23) and Eq. (24) into Eq. (22). $\qquad\square$

The following lemma is a counterpart of Lemma A.2, which shows that

$$\mathbb{E}_{(s,a)\sim d_{\widehat{P}_k}^\pi}\left[g(s,a)\right] \lesssim \mathbb{E}_{(s,a)\sim d_{\widehat{P}_k}^\pi}\left[\left\|\widehat{\phi}_k(s,a)\right\|_{\Sigma_{\rho_k\times\bar{\pi}_k,\widehat{\phi}_k}^{-1}}\right] ,$$

if $\mathbb{E}_{s\sim\rho_k',a\sim\bar{\pi}_k}\left[g^2(s,a)\right]$ is upper bounded. Note that compared with Lemma A.2, this lemma additionally needs to condition on the event that the MLE guarantee (*cf.*, Lemma C.1) holds.

**Lemma A.3** (One-step-back inequality in the learned model). *Conditioned on the event where the MLE guarantee in Lemma C.1 holds, i.e., $\mathbb{E}_{s\sim\rho_{k_i},a\sim\bar{\pi}_{k_i}}\left[f_{k_i}(s,a)^2\right] \lesssim \zeta_{k_i}$, for any epoch $i \in [N]$. Then for any $g : \mathcal{S} \times \mathcal{A} \to \mathbb{R}$ such that $\|g\|_\infty \leq B$, any epoch $i \in [N]$ and any policy $\pi$, it holds that*

$$\mathbb{E}_{(\tilde{s},\tilde{a})\sim d_{\widehat{P}_{k_i}}^\pi,s\sim\widehat{P}_{k_i}(\cdot|\tilde{s},\tilde{a}),a\sim\pi(s)}\left[g(s,a)\right]$$

$$\leq \mathbb{E}_{(\tilde{s},\tilde{a})\sim d_{\widehat{P}_{k_i}}^\pi}\left[\left\|\widehat{\phi}_{k_i}(\tilde{s},\tilde{a})\right\|_{\Sigma_{\rho_{k_i}\times\bar{\pi}_{k_i},\widehat{\phi}_{k_i}}^{-1}}\right]\sqrt{\frac{k_i A}{\xi}\mathbb{E}_{s\sim\rho_{k_i}',a\sim\bar{\pi}_{k_i}}\left[g^2(s,a)\right] + B^2\lambda_{k_i}d + k_i B^2\zeta_{k_i}} ,$$

*where recall that $\bar{\pi}_k(\cdot \mid s) = \xi \cdot U(\mathcal{A}) + (1-\xi) \cdot 1/k \sum_{j=1}^k \tilde{\pi}_j(\cdot \mid s)$.*

*Proof.* The proof of this lemma is generally similar to that of Lemma A.2. We start by applying the Cauchy–Schwarz inequality:

$$\mathbb{E}_{(\tilde{s},\tilde{a})\sim d_{\widehat{P}_{k_i}}^\pi,s\sim\widehat{P}_{k_i}(\cdot|\tilde{s},\tilde{a}),a\sim\pi(s)}\left[g(s,a)\right]$$

$$=\mathbb{E}_{(\tilde{s},\tilde{a})\sim d^{\pi}_{\widehat{P}_{k_i}}}\left[\widehat{\phi}_{k_i}(\tilde{s},\tilde{a})^{\top}\int\sum_{a}\widehat{\mu}_{k_i}(s)\pi(a\mid s)g(s,a)d(s)\right]$$

$$\leq\mathbb{E}_{(\tilde{s},\tilde{a})\sim d^{\pi}_{\widehat{P}_{k_i}}}\left[\left\|\widehat{\phi}_{k_i}(\tilde{s},\tilde{a})\right\|_{\Sigma^{-1}_{\rho_{k_i}\times\bar{\pi}_{k_i},\widehat{\phi}_{k_i}}}\left\|\int\sum_{a}\widehat{\mu}_{k_i}(s)\pi(a\mid s)g(s,a)d(s)\right\|_{\Sigma_{\rho_{k_i}\times\bar{\pi}_{k_i},\widehat{\phi}_{k_i}}}\right]. \quad (25)$$

We now bound the second quadratic form w.r.t. $\Sigma_{\rho_{k_i}\times\bar{\pi}_{k_i},\widehat{\phi}_{k_i}}$ in Eq. (25) as follows:

$$\left\|\int\sum_{a}\widehat{\mu}_{k_i}(s)\pi(a\mid s)g(s,a)d(s)\right\|^2_{\Sigma_{\rho_{k_i}\times\bar{\pi}_{k_i},\widehat{\phi}_{k_i}}}$$

$$=\left[\int\sum_{a}\widehat{\mu}_{k_i}(s)\pi(a\mid s)g(s,a)d(s)\right]^{\top}\cdot\left\{k_i\mathbb{E}_{s\sim\rho_{k_i},a\sim\bar{\pi}_{k_i}}\left[\widehat{\phi}_{k_i}(s,a)\widehat{\phi}_{k_i}(s,a)^{\top}\right]+\lambda_{k_i}I\right\}$$

$$\cdot\left[\int\sum_{a}\widehat{\mu}_{k_i}(s)\pi(a\mid s)g(s,a)d(s)\right]$$

$$\leq k_i\mathbb{E}_{\tilde{s}\sim\rho_{k_i},\tilde{a}\sim\bar{\pi}_{k_i}}\left\{\left[\int\sum_{a}\widehat{\mu}_{k_i}(s)^{\top}\widehat{\phi}_{k_i}(\tilde{s},\tilde{a})\pi(a\mid s)g(s,a)d(s)\right]^2\right\}+B^2\lambda_{k_i}d$$

$$=k_i\mathbb{E}_{\tilde{s}\sim\rho_{k_i},\tilde{a}\sim\bar{\pi}_{k_i}}\left\{\mathbb{E}_{s\sim\widehat{P}_{k_i}(\cdot|\tilde{s},\tilde{a}),a\sim\pi(s)}[g(s,a)]^2\right\}+B^2\lambda_{k_i}d\,, \quad (26)$$

where the first inequality is because $\|g(s,a)\|_{\infty}\leq B$ as well as the regularity condition in Assumption 2.1 that $\left\|\int\mu(s)h(s)\mathrm{d}(s)\right\|_2\leq\sqrt{d}$ for any $h:\mathcal{S}\to[0,1]$ and any $\mu\in\Psi$.

Moreover, using the MLE guarantee in Lemma C.1, we have that

$$k_i\mathbb{E}_{\tilde{s}\sim\rho_{k_i},\tilde{a}\sim\bar{\pi}_{k_i}}\left\{\mathbb{E}_{s\sim\widehat{P}_{k_i}(\cdot|\tilde{s},\tilde{a}),a\sim\pi(s)}[g(s,a)]^2\right\}+B^2\lambda_{k_i}d$$

$$\leq k_i\mathbb{E}_{\tilde{s}\sim\rho_{k_i},\tilde{a}\sim\bar{\pi}_{k_i}}\left\{\mathbb{E}_{s\sim P^{\star}(\cdot|\tilde{s},\tilde{a}),a\sim\pi(s)}[g(s,a)]^2\right\}+B^2\lambda_{k_i}d+k_iB^2\zeta_{k_i}$$

$$\leq k_i\mathbb{E}_{\tilde{s}\sim\rho_{k_i},\tilde{a}\sim\bar{\pi}_{k_i},s\sim P^{\star}(\cdot|\tilde{s},\tilde{a}),a\sim\pi(s)}\left[g^2(s,a)\right]+B^2\lambda_{k_i}d+B^2k_i\zeta_{k_i}$$

$$\leq\frac{k_iA}{\xi}\mathbb{E}_{\tilde{s}\sim\rho_{k_i},\tilde{a}\sim\bar{\pi}_{k_i},s\sim P^{\star}(\cdot|\tilde{s},\tilde{a}),a\sim\bar{\pi}_{k_i}}\left[g^2(s,a)\right]+B^2\lambda_{k_i}d+B^2k_i\zeta_{k_i}$$

$$\leq\frac{k_iA}{\xi}\mathbb{E}_{s\sim\rho'_{k_i},a\sim\bar{\pi}_{k_i}}\left[g^2(s,a)\right]+B^2\lambda_{k_i}d+B^2k_i\zeta_{k_i}\,, \quad (27)$$

where the second inequality follows by Jensen's inequality, the third inequality comes from importance sampling and the definition of $\bar{\pi}_{k_i}$, and the last inequality is due to the definition of $\rho'_{k_i}$.

The proof is now concluded by substituting Eq. (26) and Eq. (27) into Eq. (25). $\qquad\square$

The following lemma shows that the expectation of any state-action function $g:\mathcal{S}\times\mathcal{A}\to\mathbb{R}$ w.r.t. $d^{\pi_1}_P$ and $\pi_2$ can be decomposed into (a) the one-step-back expectation of $g$ w.r.t. $d^{\pi_1}_P$ and $\pi_2$; and (b) the expectation of $g$ w.r.t. $d_0$ and $\pi_2$.

**Lemma A.4.** *For any $P$, and any policy $\pi_1$ and $\pi_2$, it holds that*

$$\mathbb{E}_{s\sim d^{\pi_1}_P,a\sim\pi_2(\cdot|s)}[g(s,a)]$$
$$=\gamma\mathbb{E}_{(\tilde{s},\tilde{a})\sim d^{\pi_1}_P,s\sim P(\cdot|\tilde{s},\tilde{a}),a\sim\pi_2(\cdot|s)}[g(s,a)]+(1-\gamma)\mathbb{E}_{s\sim d_0,a\sim\pi_2(\cdot|s_0)}[g(s,a)]\,.$$

*Proof.*

$$\mathbb{E}_{s\sim d^{\pi_1}_P,a\sim\pi_2(\cdot|s)}[g(s,a)]$$
$$=\sum_{t=1}^{\infty}(1-\gamma)\gamma^t\mathbb{E}_{s\sim d^{\pi_1}_{P,t},a\sim\pi_2(\cdot|s)}[g(s,a)]+(1-\gamma)\mathbb{E}_{s\sim d^{\pi_1}_{P,0},a\sim\pi_2(\cdot|s)}[g(s,a)]$$

$$
\begin{aligned}
=& \gamma \sum_{t=0}^{\infty}(1-\gamma)\gamma^t \mathbb{E}_{s \sim d_{P,t+1}^{\pi_1}, a \sim \pi_2(\cdot|s)}[g(s,a)] + (1-\gamma)\mathbb{E}_{s \sim d_0, a \sim \pi_2(\cdot|s_0)}[g(s,a)] \\
=& \gamma \sum_{t=0}^{\infty}(1-\gamma)\gamma^t \mathbb{E}_{(\tilde{s},\tilde{a}) \sim d_{P,t}^{\pi_1}, s \sim P(\cdot|\tilde{s},\tilde{a}), a \sim \pi_2(\cdot|s)}[g(s,a)] + (1-\gamma)\mathbb{E}_{s \sim d_0, a \sim \pi_2(\cdot|s_0)}[g(s,a)] \\
=& \gamma \mathbb{E}_{(\tilde{s},\tilde{a}) \sim d_P^{\pi_1}, s \sim P(\cdot|\tilde{s},\tilde{a}), a \sim \pi_2(\cdot|s)}[g(s,a)] + (1-\gamma)\mathbb{E}_{s \sim d_0, a \sim \pi_2(\cdot|s_0)}[g(s,a)].
\end{aligned}
$$

$\square$

# B  Omitted Analysis of The Regret Lower Bound

In this section, we provide the proof of Theorem 4.2. For the remainder of this section, we switch from loss functions to reward functions for convenience since we now consider MDPs with fixed loss functions.

## B.1  Construction of Hard-to-learn MDP Instances

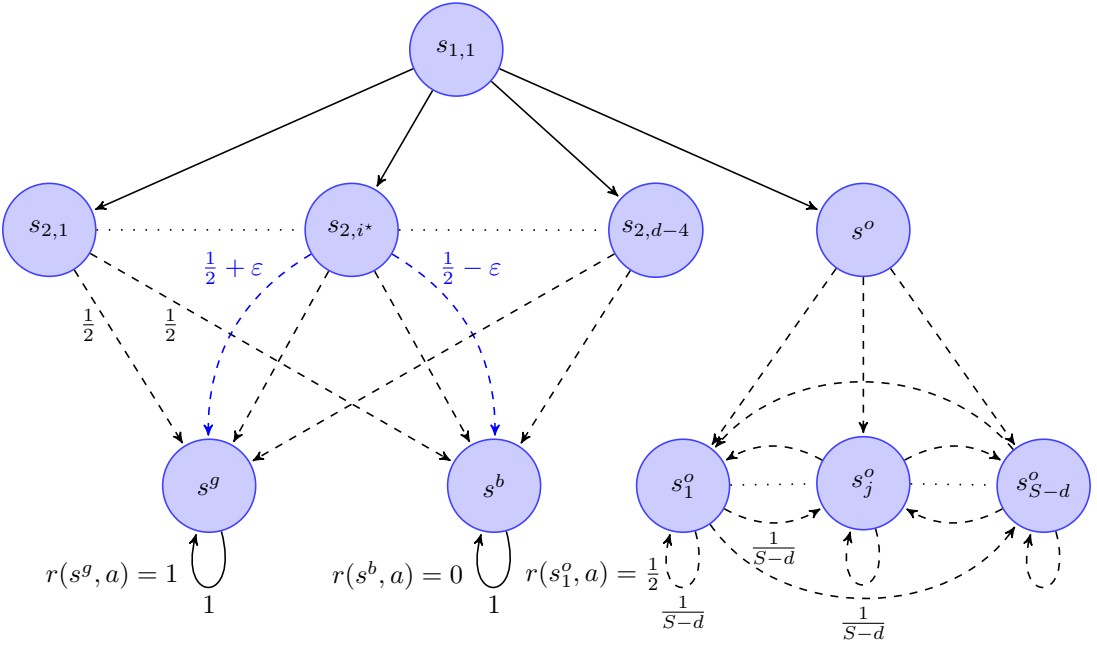

Figure 1: The class of the hard-to-learn low-rank MDP instances used in the proof of Theorem 4.2.

To prove our regret lower bound in Theorem 4.2, we construct a class of hard-to-learn low-rank MDP instances, as shown in Figure 1. Note that similar hard MDP instances are first introduced to prove the regret lower bounds for tabular MDPs [Lattimore and Szepesvári, 2020, Domingues et al., 2021] and are recently also used to prove the lower bound of sample complexity for learning low-rank MDPs by Cheng et al. [2023].

To begin with, we first introduce the reference low-rank MDP $\mathcal{M}_0$, with its elements detailed as follows:

- State space: $\mathcal{S} = \{s_{1,1}, s_{2,1}, s_{2,2}, \ldots, s_{2,d-4}\} \cup \{s^g, s^b\} \cup \{s^o\} \cup \mathcal{S}_{\mathcal{O}}$, where $\mathcal{S}_{\mathcal{O}} = \{s_i^o\}_{i=1}^{S-d}$ denotes the set of "outlier states", $s^g$ denotes the "good state", and $s^b$ denotes the "bad state".
- Action space: $\mathcal{A} = \{a_1, a_2, \ldots, a_A\}$.
- Reward function: $r(s,a) = \mathbb{I}\{s = s^g\} + \frac{1}{2}\mathbb{I}\{s \in \mathcal{S}_{\mathcal{O}}\}$.

- Transitions:
    - For the initial state $s_{1,1}$, the learner will deterministically transit to state $s_{2,i}$ if taking action $a_i$, $\forall i \in [d-4]$, and will transit to state $s^o$ otherwise. Formally, $P^\star(s_{2,i} \mid s_{1,1}, a_i) = 1$, $\forall i \in [d-4]$, and $P^\star(s^o \mid s_{1,1}, a_i) = 1$, $\forall i \in [A] \setminus [d-4]$.
    - For state $s_{2,i} \in \{s_{2,1}, s_{2,2}, \ldots, s_{2,d-4}\}$, the learner will transit to good state $s^g$ and bad state $s^b$ uniformly at random, no matter what action it takes, *i.e.*, $P^\star(s^g \mid s_{2,i}, a) = P^\star(s^b \mid s_{2,i}, a) = \frac{1}{2}$, $\forall i \in [d-4]$ and $a \in \mathcal{A}$.
    - For states $s^o$ and $s_i^o \in \mathcal{S}_{\mathcal{O}}$, the learner will uniformly transit to a state $s_j^o \in \mathcal{S}_{\mathcal{O}}$, no matter what action it takes. Formally, $P^\star(s_j^o \mid s^o, a) = P^\star(s_j^o \mid s_i^o, a) = \frac{1}{S-d}$, $\forall s_i^o, s_j^o \in \mathcal{S}_{\mathcal{O}}$ and $a \in \mathcal{A}$.
    - For states $s^g$ and $s^b$, the learner will stay at the current state no matter what action it takes, which means that $P^\star(s^g \mid s^g, a) = P^\star(s^b \mid s^b, a) = 1$.

Further, the transitions of the above MDP can be realized by $P^\star(s' \mid s, a) = \langle \phi^\star(s,a), \mu^\star(s') \rangle$, with the following features, which thus implies that this MDP is indeed a low-rank MDP:

$$\phi^\star(s_{1,1}, a_i) = \boldsymbol{e}_i, \quad \mu^\star(s_{2,i}) = \boldsymbol{e}_i, \ \forall i \in [d-4], \quad \mu^\star(s_{1,1}) = \boldsymbol{0}$$

$$\phi^\star(s_{1,1}, a_i) = (0, \ldots, 0, 1, 0), \ \forall i \in [A] \setminus [d-4], \quad \mu^\star(s^o) = (0, \ldots, 0, 1, 0)$$

$$\phi^\star(s_{2,j}, a) = (0, \ldots, 0, \frac{1}{2}, \frac{1}{2}, 0, 0), \ \forall a \in \mathcal{A}, \quad \mu^\star(s^g) = (0, \ldots, 0, 1, 0, 0, 0), \quad \mu^\star(s^b) = (0, \ldots, 0, 0, 1, 0, 0)$$

$$\phi^\star(s^o, a) = \phi^\star(s_j^o, a) = (0, \ldots, 0, \frac{1}{S-d}), \ \forall a \in \mathcal{A}, \quad \mu^\star(s_j^o) = (0, \ldots, 0, 1)$$

$$\phi^\star(s^g, a) = \mu^\star(s^g), \quad \phi^\star(s^b, a) = \mu^\star(s^b), \ \forall a \in \mathcal{A}.$$

Based on the reference MDP $\mathcal{M}_0$, we define other low-rank MDP instances $\mathcal{M}_{(i^\star, a^\star)}$, $\forall (i^\star, a^\star) \in [d-4] \times \mathcal{A}$. In specific, the only difference between $\mathcal{M}_{(i^\star, a^\star)}$ and $\mathcal{M}_0$ is that $\phi^\star(s_{2,i^\star}, a^\star) = (0, \ldots, 0, \frac{1}{2} + \varepsilon, \frac{1}{2} - \varepsilon, 0, 0)$, such that $P^\star(s_g \mid s_{2,i^\star}, a^\star) = \frac{1}{2} + \varepsilon$, and $P^\star(s_b \mid s_{2,i^\star}, a^\star) = \frac{1}{2} - \varepsilon$, for some $\varepsilon > 0$ to be defined later.

## B.2 Proof of Theorem 4.2

Based on the class of hard-to-learn low-rank MDP instances constructed above, we are now ready to prove the regret lower bound in Theorem 4.2.

*Proof of Theorem 4.2.* In what follows, we denote by $\mathbb{P}_{(i^\star, a^\star)} := \mathbb{P}_{\text{Alg}, \mathcal{M}_{(i^\star, a^\star)}}$ the probability measure over the outcomes induced by the interaction between Alg and $\mathcal{M}_{(i^\star, a^\star)}$, and by $\mathbb{E}_{(i^\star, a^\star)} := \mathbb{E}_{\text{Alg}, \mathcal{M}_{(i^\star, a^\star)}}$ the expectation with respect to $\mathbb{P}_{(i^\star, a^\star)}$.

**Regret of** Alg **in** $\mathcal{M}_{(i^\star, a^\star)}$   For some $\mathcal{M}_{(i^\star, a^\star)}$, its optimal policy $\pi^\star_{(i^\star, a^\star)} : \mathcal{S} \to \mathcal{A}$ satisfies that $\pi^\star_{(i^\star, a^\star)}(s_{1,1}) = a_{i^\star}$ and $\pi^\star_{(i^\star, a^\star)}(s_{2,i^\star}) = a^\star$, with the optimal value function

$$V_0^\star(s_{1,1}) = \mathbb{E}\left[\sum_{\tau=0}^{+\infty} \gamma^\tau r(s_\tau, a_\tau) \mid \pi^\star_{(i^\star, a^\star)}, P^\star_{(i^\star, a^\star)}, s_0 = s_{1,1}\right] = \sum_{\tau=2}^{+\infty} \gamma^\tau \left(\frac{1}{2} + \varepsilon\right) = \frac{\gamma^2}{1-\gamma}\left(\frac{1}{2} + \varepsilon\right). \tag{28}$$

For some policy $\pi$, it is also clear that its value function satisfies

$$V_0^\pi(s_{1,1}) = \frac{\gamma^2}{1-\gamma}\left(\frac{1}{2} + \varepsilon \mathbb{P}_{(i^\star, a^\star)}((s_2, a_2) = (s_{2,i^\star}, a^\star))\right). \tag{29}$$

Combining Eq. (28) and (29) shows that the regret of Alg in $\mathcal{M}_{(i^\star, a^\star)}$ satisfies

$$\mathcal{R}_K(\text{Alg}, \mathcal{M}_{(i^\star, a^\star)}) = \frac{\gamma^2 \varepsilon}{1-\gamma} K \left(1 - \frac{1}{K} \mathbb{E}_{(i^\star, a^\star)}\left[\sum_{k=1}^K \mathbb{I}\{(s_2^k, a_2^k) = (s_{2,i^\star}, a^\star)\}\right]\right)$$

$$= \frac{\gamma^2 \varepsilon}{1-\gamma} K \left(1 - \frac{1}{K} \mathbb{E}_{(i^\star, a^\star)}\left[N_{(i^\star, a^\star)}^K\right]\right),$$

where we define $N_{(i^\star, a^\star)}^K := \sum_{k=1}^K \mathbb{I}\{(s_2^k, a_2^k) = (s_{2,i^\star}, a^\star)\}$.

**Maximum Regret of** Alg **over All Possible** $\mathcal{M}_{(i^\star,a^\star)}$  With $\mathcal{R}_K(\text{Alg},\mathcal{M}_{(i^\star,a^\star)})$ in the above equation, we can deduce that

$$
\begin{aligned}
\max_{(i^\star,a^\star)} \mathcal{R}_K(\text{Alg},\mathcal{M}_{(i^\star,a^\star)}) &\geq \frac{1}{(d-4)A} \sum_{(i^\star,a^\star)} \mathcal{R}_K(\text{Alg},\mathcal{M}_{(i^\star,a^\star)}) \\
&\geq \frac{\gamma^2 \varepsilon}{1-\gamma} K \left( 1 - \frac{1}{K(d-4)A} \sum_{(i^\star,a^\star)} \mathbb{E}_{(i^\star,a^\star)} \left[ N_{(i^\star,a^\star)}^K \right] \right) . \quad (30)
\end{aligned}
$$

To lower bound the above display, it remains to upper bound $\sum_{(i^\star,a^\star)} \mathbb{E}_{(i^\star,a^\star)} \left[ N_{(i^\star,a^\star)}^K \right]$. To this end, by Lemma 1 in the work of Garivier et al. [2019] together with the fact that $N_{(i^\star,a^\star)}^K / K \in [0,1]$, it holds that

$$
\text{KL}\left( \text{Ber}\left( \frac{1}{K} \mathbb{E}_0 \left[ N_{(i^\star,a^\star)}^K \right] \right), \text{Ber}\left( \frac{1}{K} \mathbb{E}_{(i^\star,a^\star)} \left[ N_{(i^\star,a^\star)}^K \right] \right) \right) \leq \text{KL}\left( \mathbb{P}_0, \mathbb{P}_{(i^\star,a^\star)} \right) .
$$

This implies that

$$
\begin{aligned}
\frac{1}{K} \mathbb{E}_{(i^\star,a^\star)} \left[ N_{(i^\star,a^\star)}^K \right] &\leq \frac{1}{K} \mathbb{E}_0 \left[ N_{(i^\star,a^\star)}^K \right] + \sqrt{\frac{1}{2} \text{KL}\left( \mathbb{P}_0, \mathbb{P}_{(i^\star,a^\star)} \right)} \\
&= \frac{1}{K} \mathbb{E}_0 \left[ N_{(i^\star,a^\star)}^K \right] + \varepsilon \sqrt{2} \sqrt{\mathbb{E}_0 \left[ N_{(i^\star,a^\star)}^K \right]},
\end{aligned}
$$

where the inequality is due to Pinsker's inequality that $(p-q)^2 \leq \frac{1}{2} \text{KL}(\text{Ber}(p),\text{Ber}(q))$, for $p,q \in [0,1]$, and the equality comes from Lemma 15.1 of Lattimore and Szepesvári [2020] and Lemma 14 of Domingues et al. [2021] as well as assuming $0 \leq \varepsilon \leq \frac{1}{4}$.

Based on this, one can see that

$$
\begin{aligned}
\frac{1}{K} \sum_{(i^\star,a^\star)} \mathbb{E}_{(i^\star,a^\star)} \left[ N_{(i^\star,a^\star)}^K \right] &\leq \frac{1}{K} \sum_{(i^\star,a^\star)} \mathbb{E}_0 \left[ N_{(i^\star,a^\star)}^K \right] + \varepsilon \sqrt{2} \sum_{(i^\star,a^\star)} \sqrt{\mathbb{E}_0 \left[ N_{(i^\star,a^\star)}^K \right]} \\
&\leq 1 + \varepsilon \sqrt{2} \sqrt{(d-4)AK}, \quad (31)
\end{aligned}
$$

where the second inequality follows from using the Cauchy-Schwartz inequality together with the fact that $N_{(i^\star,a^\star)}^K \leq K$.

**Optimizing $\varepsilon$ to Lower Bound the Maximum Regret**  Substituting Eq. (31) into Eq. (30) leads to

$$
\begin{aligned}
\max_{(i^\star,a^\star)} \mathcal{R}_K(\text{Alg},\mathcal{M}_{(i^\star,a^\star)}) &\geq \frac{\gamma^2 \varepsilon}{1-\gamma} K \left( 1 - \frac{1}{(d-4)A} - \varepsilon \sqrt{2} \sqrt{\frac{K}{(d-4)A}} \right) \\
&\geq \frac{1}{4\sqrt{2}} \cdot \frac{\gamma^2}{1-\gamma} \left( 1 - \frac{1}{(d-4)A} \right)^2 \sqrt{(d-4)AK} \\
&\geq \frac{361}{1600\sqrt{2}} \cdot \frac{\gamma^2}{1-\gamma} \sqrt{(d-4)AK},
\end{aligned}
$$

where the second inequality comes from by choosing $\varepsilon = \frac{1}{2\sqrt{2}} \left( 1 - \frac{1}{(d-4)A} \right) \sqrt{\frac{(d-4)A}{K}}$ and the last inequality is due to $d \geq 8$ and $A \geq d-3$. Finally, note that $\varepsilon \leq \frac{1}{4}$ is guaranteed when $K \geq 2(d-4)A$. The proof is thus concluded. $\qquad\square$

## C  Auxiliary Lemmas

We first introduce the concentration of MLE, the i.i.d. version of which at least dates back to Chapter 7 of Geer [2000] and the non-i.i.d. version of which is first proved by Agarwal et al. [2020] and also appears in the analysis of Uehara et al. [2022].

**Lemma C.1** (MLE guarantee). *For some fixed epoch $i \in [N]$, with probability $1 - \delta$, it holds that*

$$\mathbb{E}_{s \sim \left\{ 0.5\rho_{k_i} + 0.5\rho'_{k_i} \right\}, a \sim \bar{\pi}_{k_i}(s)} \left[ \left\| \widehat{P}_{k_i}(\cdot \mid s, a) - P^{\star}(\cdot \mid s, a) \right\|_1^2 \right] \lesssim \zeta, \quad \zeta := \frac{\ln(M/\delta)}{k_i}.$$

*Therefore, simultaneously for all epoch $i \in [N]$, with probability $1 - \delta$, it holds that*

$$\mathbb{E}_{s \sim \left\{ 0.5\rho_{k_i} + 0.5\rho'_{k_i} \right\}, a \sim \bar{\pi}_{k_i}(s)} \left[ \left\| \widehat{P}_{k_i}(\cdot \mid s, a) - P^{\star}(\cdot \mid s, a) \right\|_1^2 \right] \lesssim \zeta_{k_i}, \quad \zeta_{k_i} := \frac{\ln(MN/\delta)}{k_i}.$$

The following lemma is the canonical elliptical potential lemma.

**Lemma C.2** (Lemma 19.4, Lattimore and Szepesvári [2020]). *Let $M_0 = \lambda_0 I \in \mathbb{R}^{d \times d}$ with $\lambda_0 > 0$ and $M_k = M_{k-1} + G_k$, where $G_k$ is positive definite with the maximum eigenvalue $\lambda_{\max}(G_k) \leq 1$ and $\mathrm{Tr}(G_k) \leq B^2$. Then*

$$\sum_{k=1}^{K} \mathrm{Tr}(G_k M_{k-1}^{-1}) \leq 2 \ln \det(M_K) - 2 \ln \det(M_0) \leq 2d \ln \left( 1 + \frac{KB^2}{d\lambda_0} \right).$$

The following lemma guarantees the concentration of the empirical feature covariance matrix and the version for fixed feature mapping $\phi(\cdot)$ is first proved by Zanette et al. [2021]. The proof of this lemma can be readily obtained by taking a union bound over any $\phi \in \Phi$ in the proof of Lemma 39 of Zanette et al. [2021].

**Lemma C.3.** *Let $\lambda_{k_i} = \Theta(d \ln(k_i |\Phi|/\delta)) = \Theta(d \ln(k_i M/\delta)), \forall i \in [N]$. Then simultaneously for all $i \in [N]$ and all $\phi \in \Phi$, with probability $1 - \delta$, it holds that*

$$\|\phi(s, a)\|_{\widehat{\Sigma}_{k_i, \phi}^{-1}} = \Theta \left( \|\phi(s, a)\|_{\Sigma_{\rho_{k_i} \times \bar{\pi}_{k_i}, \phi}^{-1}} \right).$$

The following is the canonical simulation lemma, which bounds the difference between the performance of the same policy $\pi$ under two different environments and dates back at least to Abbeel and Ng [2005].

**Lemma C.4** (Simulation lemma). *Given two MDP models $(P', \ell - b)$ and $(P, \ell)$, for any policy $\pi$, it holds that*

$$V_{P',\ell-b}^{\pi} - V_{P,\ell}^{\pi} = \frac{1}{1-\gamma} \mathbb{E}_{(s,a) \sim d_{P'}^{\pi}} \left[ -b(s, a) + \gamma \left( P'(\cdot \mid s, a) - P(\cdot \mid s, a) \right)^{\top} V_{P,\ell}^{\pi} \right], \qquad (32)$$

*and*

$$V_{P',\ell-b}^{\pi} - V_{P,\ell}^{\pi} = \frac{1}{1-\gamma} \mathbb{E}_{(s,a) \sim d_{P}^{\pi}} \left[ -b(s, a) + \gamma \left( P'(\cdot \mid s, a) - P(\cdot \mid s, a) \right)^{\top} V_{P',\ell-b}^{\pi} \right]. \qquad (33)$$