# OpenReview forum: "Learning Adversarial Low-rank Markov Decision Processes with Unknown Transition and Full-information Feedback"
_NeurIPS.cc/2023/Conference — NeurIPS 2023 poster_

### Official Review · Reviewer_aWbZ · 2023-07-02

**Soundness:** 3 good
**Presentation:** 3 good
**Contribution:** 2 fair
**Rating:** 6
**Confidence:** 3

**Summary:**

The paper studies low-rank MDPs with adversarially changing losses in the full-information feedback setting. They assume the unknown transition probability function admits a low-rank matrix decomposition. They present POLO algorithm, a policy optimization-based algorithm, and prove it has sublinear regret in the number of episodes $K$, but linear dependency in $d$ which is the rank of the transition kernel. The regret is w.r.t the optimal policy for the discounted return value function.
The authors claim they are the first to present and algorithm that interleaves representation learning, exploration, and exploitation to achieve the sublinear regret  guarantee for RL with nonlinear function approximation and adversarial losses.


**Strengths:**

1.	POLO is the first algorithm that achieves sublinear regret with unknown features (although the setting is full information).
2.	The authors provide a clear comparison between their work to previous works in the area, which emphasis their contribution.
3.     Regret upper bound stated clearly, the analysis sketch is standard and clear (not fully checked).


**Weaknesses:**

1.	The full information feedback is a restrictive and unrealistic assumption.
2.	The extensive literature review in Section 2 can be shorter, and instead use that space to have more proof sketches. That, in my opinion, can benefit the reader.


**Questions:**

1.	What is the difference between low-rank MDPs to Generalized Linear Model? As I see it, GLM is more general model. Do your results hold for this setting as well?
2.	Same question for low Bellman rank MDPs, that also seem to generalize low-rank MDPs.
3.	Do you have thoughts about how to remove the full-information assumption?

---

> ### Author Rebuttal · Authors · 2023-08-10
>
> We thank the reviewer for the valuable comments and suggestions. Our response to each question is provided in turn below.
>
> **Q1. "The full information feedback is a restrictive and unrealistic assumption."**
>
> We do believe that extending the analysis of our work to the bandit feedback case is an interesting and important next step, as mentioned in Section 6. However, we would like to note that, even if in linear mixture MDPs with given true feature mappings, it is also common to first study adversarial environments with full-information feedback [1,2], before moving to the bandit feedback case [3]. Moreover, the SOTA regret guarantee obtained by policy optimization (PO) based methods for the more amenable adversarial linear MDPs with given true features in the full-information setting is $\widetilde{O}(K^{3/4})$ (omitting other dependence), as shown by a concurrent work [4]. Our algorithm obtains the regret guarantee with the same dependence on $K$ and can additionally work when no true feature mappings are known a priori. Besides, we would like to remark that, learning adversarial linear MDPs with bandit feedback is already rather challenging even with the true feature mappings given, where the SOTA regret is only of $\widetilde{O}(K^{6/7})$ order [5] (to the best of our knowledge) when no simulators or exploratory assumptions are given.
>
> **Q2. "The extensive literature review in Section 2 can be shorter, ... can benefit the reader."**
>
> We will revise our paper accordingly to make Section 2 more succinct and include more proof sketches for better readability. In specific, we plan to (a) merge Section 2.1 and 2.2 into a new section; and (b) include more proof sketches and discussions of our Theorem 5.1 and related key lemmas in Section 5.
>
> **Q3. "What is the difference between low-rank MDPs to Generalized Linear Model? ..."**
>
> In GLMs, it is assumed that $\mathbb{E}[Y \mid X]=\mu\left(X^\top \theta^\star\right)$, where $\theta^\star \in \mathbb{R}^d$ is the unknown parameter, $X$ is the **known** feature vector, $Y$ is the response variable, and $\mu: \mathbb{R} \rightarrow \mathbb{R}$ is the link function. On the other hand, in low-rank MDPs, the transition $P^\star$ is assumed to admit a low-rank decomposition, meaning that $\mathbb{E}[\mathbb{I}\{s=s_{t+1}\} \mid s_t,a_t]=\mu^{\star}\left(s\right)^{\top} \phi^{\star}(s_t, a_t)$, where both the feature vectors $\mu^{\star}\left(s\right)$and $ \phi^{\star}(s_t, a_t)$ are **unknown**. Therefore, in general, these two models can not imply each other and the results for each model might not be directly comparable.
>
> **Q4. "Same question for low Bellman rank MDPs, ..."**
>
> Indeed, there exist some other works studying RL with general function approximation [6,7,8], which subsumes low-rank MDPs as a special case. However, we note that the algorithms in these works are version space algorithms and thus are generally not computationally efficient. In contrast, the algorithms in previous works studying low-rank MDPS including ours are computationally feasible in practice, as discussed in our Section 7. Besides, they assume deterministic reward functions, and thus their results are not applicable in our setting.
>
> **Q5. "Do you have thoughts about how to remove the full-information assumption?"**
>
> To tackle bandit feedback, if we restrict our attention to PO-based methods, one may consider the *dilated* bonuses to facilitate improved global exploration, which is vital for current PO-based methods to learn adversarial linear MDPs with bandit feedback [9,10,5]. Nevertheless, the construction of such dilated bonuses critically relies on the unknown true feature mapping $\phi^\star(\cdot,\cdot)$, and thus it is currently not clear whether they can be extended to low-rank MDPs. Also, as aforementioned, the SOTA regret of such methods to learn adversarial linear MDPs with bandit feedback is only of $\widetilde{O}(K^{6/7})$ order [5]. Besides, if occupancy measure-based methods are also under consideration, the feasible way might be constructing some sort of confidence set for the learned empirical transitions, like the confidence intervals constructed in tabular MDPs [11] and the ellipsoid confidence sets constructed in linear mixture MDPs [3], to construct optimistically biased loss estimates to tackle bandit feedback. However, as mentioned in Section 6, it is also not very clear how to guarantee such point-wise optimism at each state-action pair in low-rank MDPs. Overall, learning adversarial low-rank MDPs with bandit feedback is indeed interesting and also particularly challenging, and we believe our work may serve as an important step to further tackle this problem, which we leave as our future study.
>
>
> [1] Cai et al. Provably Efficient Exploration in Policy Optimization. ICML, 20.
>
> [2] He et al. Near-optimal Policy Optimization Algorithms for Learning Adversarial Linear Mixture MDPs. AISTATS, 22.
>
> [3] Zhao et al. Learning adversarial linear mixture markov decision processes with bandit feedback and unknown transition. ICLR, 23.
>
> [4] Zhong et al. A Theoretical Analysis of Optimistic Proximal Policy Optimization in Linear Markov Decision Processes. arXiv, 23.
>
> [5] Sherman et al. Improved Regret for Efficient Online Reinforcement Learning with Linear Function Approximation. ICML, 23.
>
> [6] Jiang et al. Contextual decision processes with low bellman rank are pac-learnable. ICML, 17.
>
> [7] Sun et al. Model-based rl in contextual decision processes: Pac bounds and exponential improvements over model-free approaches. COLT, 19.
>
> [8] Du et al. Bilinear classes: A structural framework for provable generalization in rl. ICML, 2021.
>
> [9] Luo et al. Policy optimization in adversarial mdps: Improved exploration via dilated bonuses. NeurIPS, 21.
>
> [10] Dai et al. Refined Regret for Adversarial MDPs with Linear Function Approximation. ICML, 23.
>
> [11] Jin et al. Learning Adversarial Markov Decision Processes with Bandit Feedback and Unknown Transition. ICML, 20.

---

> > ### Comment · Reviewer_aWbZ · 2023-08-10
> >
> > I thank the authors for their response and have no further questions.

---

> > > ### Author Response · Authors · 2023-08-11
> > > **Author Response**
> > >
> > > Thank you for your response to our rebuttal! If there are any additional questions or comments, we would also be more than happy to address them.

---

### Official Review · Reviewer_ci4p · 2023-07-04

**Soundness:** 3 good
**Presentation:** 3 good
**Contribution:** 3 good
**Rating:** 6
**Confidence:** 3

**Summary:**

This work focuses on the low-rank MDPs with adversarial losses in the full-information feedback setting. Different from the previous work which assumes known features, this work considers the combination of representation learning and regret minimization problem, and is of the first result under this specific topic.

The low-rank MDPs are defined in Definition 3.1, which admit certain low-rank feature structure that the transition dynamics can be presented with respect to two feature embedding functions. As described in the interaction protocol, the learner does not have accesses to the feature functions, and is required to learn these functions via the interaction between the environment.

While the learner has to collect enough samples to estimate the feature functions, the goal of the learner is to minimize her regret (defined in Line 201), that is, the difference between the cumulative loss suffered, and that of the optimal policy in hindsight. The loss functions are selected adversarially   that can change from episode to episode arbitrarily.

To tackle these two problems simultaneously, the authors combine the techniques of representation learning and the RL with linear MDPs and adversarial losses together into Algorithm 1. Line 5-10 follow the idea of Uehara et al. [2022] to collect the samples for representation learning, Line 11-17 follow the idea of learning adversarial Linear MDPs like the work of Cai et al. [2020], given the estimated feature functions solved in Eq. (1).

The result is presented in Theorem 5.1, which guarantees that the regret of Algorithm 1 is bounded by $\widetilde{\mathcal{O}}( \frac{K^{3/4}}{1-\gamma} + \frac{\sqrt{K}}{(1-\gamma)^2})$, where the first term is attributed to the representation learning of unknown features, and the second term is for the adversarial losses.

**Strengths:**

1. Prior to this work, it is unknown whether the regret minimization problem and the representation learning problem can be dealt simultaneously with adversarial losses. This work is the first one to achieve sub-linear regret, which sheds light on this direction.
2. Based on the previous works on the representation learning of MDPs (such as Uehara et al. [2022] and Agarwal et al. [2020]), and on the linear MDPs (or low-rank MDPs) with adversarial losses (such as Cai et al. [2020] and Luo et al. [2021a]), the authors combine the techniques of these two fields hierarchically, under a novel scheme `doubled exploration and exploitation', to address the regret minimization problem.

**Weaknesses:**

1. The designed algorithm suffers from huge computation cost (Line 13, 15, 16, and 17) when the state space is very large, which is the same as the previous algorithm OPPO in the previous work of Cai et al. [2020]. It is not very clear whether this could be improved with any planner oracle.

**Questions:**

1. This result $\mathcal{O}(K^{3/4})$ does not match the $\sqrt{K}$ regret lower bound for the adversarial losses. Any specific conjecture on the lower bound/upper bound?
2. Any conjecture on the bandit feedback setting when limited the MDPs to the linear Mixture MDPS of Zhao et al. [2023]?
3. The loss function $\ell_k$ is designed as a function of $ S \times A \rightarrow [0,1]$, which does not admit the linear structure. Any idea to improve the result given the fact that $\ell_k(s,a) = \theta_k^\top \phi^\star(s,a) $ when $\theta_k$ is revealed at the end of the episode $k$?
4. The second term in the regret bound, comparing with the regret bound of OPPO and LSUOB-REPS, shaves a factor of $d$? Could you explain why this holds true, and whether this implies that your algorithm achieves better performance given the feature functions?

**Limitations:**

This work is pure theoretical, does not have any potential negative societal impact.

---

> ### Author Rebuttal · Authors · 2023-08-09
>
> We thank the reviewer for the valuable comments and suggestions. Our response to each question is provided in turn below.
>
> **Q1. "The designed algorithm suffers from huge computation cost ...."**
>
> Indeed, our algorithm has nearly the same computation efficiency as previous works studying policy-optimization (PO) methods for RL with linear function approximation and adversarial losses [1,2], except for the MLE procedure over the experienced transitions proceeded on Line 13 of our algorithm, which might be an additional computation bottleneck due to its nonconvex nature. However, as mentioned in Section 7, the same MLE procedure is also required in previous works studying low-rank MDPs [3,4,5], arising from the fact that the underlying feature mappings are no longer known a priori in low-rank MDPs. More importantly, though being a nonconvex optimization problem, the MLE procedure can be approximately solved by gradient descent methods (*e.g.*, using neural networks), which is thus computationally feasible in practice.
>
> **Q2. "Any specific conjecture on the lower bound/upper bound?"**
>
> We conjecture that the part dealing with adversarial losses in our regret upper bound is tight. Besides, the dependence on action space size $A$ of the part in our upper bound dealing with representation learning is tight as shown by our upper bound as well as our new regret lower bounds (please see this in our response to reviewer n9xa), and we conjecture that the dependence on $\gamma$ of this part in our upper bound is tight but the dependence on $K$ might be further improved.
>
> **Q3. "Any conjecture on the bandit feedback setting when limited the MDPs to the linear Mixture MDPS of Zhao et al. [2023]?"**
>
> When learning in linear Mixture MDPS with given true feature mapping, we can construct ellipsoid confidence sets of transitions so as to construct optimistically biased loss estimates to tackle bandit feedback as in [7], but this is not aligned with our motivation to study adversarial RL where the true feature mappings are not known a priori, and thus beyond the scope of our paper.
>
> **Q4. "Any idea to improve the result given the fact that $\ell_k(s, a)=\theta_k^{\top} \phi^{\star}(s, a)$ when $\theta_k$ is revealed at the end of the episode $k$?"**
>
> Since we study adversarial low-rank MDPs with full-information feedback, imposing structural assumptions over loss functions or not actually does not affect the polynomial dependence on the number of entries of the loss functions. Indeed, due to our Lemma 5.1, the regret contributed by dealing with the adversarial loss functions does not have polynomial dependence on $S$ or $A$. In fact, as shown by, for example, Proposition 28.6 and 28.7 in [6], optimizing the online linear optimization problem with full-information feedback using OMD leads to no polynomial dependence on the number of entries of the loss functions in the unit-ball and probability simplex space, even if there are no structural assumptions imposed over the loss functions. On the other hand, in the bandit feedback case, additional structural assumptions over the loss functions are indeed needed to lift the dependence on the number of entries of the loss functions.
>
> **Q5. "The second term in the regret bound, comparing with the regret bound of OPPO and LSUOB-REPS, shaves a factor of $d$? Could you explain why this holds true, and whether this implies that your algorithm achieves better performance given the feature functions?"**
>
> Indeed, the second term in our upper bound does not depend on $d$, but this does not mean that our regret upper bound shaves a factor of $d$, compared with previous works [1,2,7]. Specifically, in adversarial environments with full-information feedback, the regret contributed by dealing with the adversarial losses (*i.e.*, the OMD Regret Term in Eq. (4) of our paper) does not have polynomial dependence on dimension $d$ as well as $S$ and $A$, as previous works studying adversarial linear mixture MDPs with full-information feedback [1,2]. For instance, the analogous term in [2] is their term $I_1$, which can be bounded by $\widetilde{O}(H^2\sqrt{K})$ as shown in their Section 6 and thus corresponds to our second term $\widetilde{O}(\sqrt{K}/(1-\gamma)^2)$. For the dependence on $d$ of regret in [1,2], it mainly comes from the difference between the values of the same policy in the learned and true models, which results from the model error and corresponds to the Estimation Bias Term in Eq. (4) of our paper. Such dependence on $d$ appears in both their bounds and the first term in our bound. For the dependence on $d$ in [7], it comes from the estimation error between the occupancy measures induced by the learned model and true model respectively, which is thus also due to the model error. In a nutshell, the fact that there is no polynomial dependence on $d$  in the second term of our regret upper bound does not conflict with the results in previous works. For the same reason, it might not be appropriate to interpret this as that our algorithm will achieve better performance given the feature functions.
>
> Thanks for pointing this out and we will include the above discussions in the revision of our paper for clarity.
>
>
>
> [1] Cai et al. Provably Efficient Exploration in Policy Optimization. ICML, 20.
>
> [2] He et al. Near-optimal Policy Optimization Algorithms for Learning Adversarial Linear Mixture MDPs. AISTATS, 22.
>
> [3] Agarwal et al. FLAMBE: structural complexity and representation learning of low rank mdp. NeurIPS, 20.
>
> [4] Uehara et al. Representation learning for online and offline RL in low-rank mdps. ICLR, 22.
>
> [5] Ni et al. Representation learning for general-sum low-rank markov games. ICLR, 23.
>
> [6] Lattimore et al. Bandit algorithms. Cambridge University Press, 20.
>
> [7] Zhao et al. Learning adversarial linear mixture markov decision processes with bandit feedback and unknown transition. ICLR, 23.

---

> > ### Comment · Reviewer_ci4p · 2023-08-17
> > **Thank the authors for their response**
> >
> > Thank the authors for their response. I would like to keep my score. Good work!

---

> > > ### Author Response · Authors · 2023-08-17
> > >
> > > Thank you for your positive feedback for our work! We are also happy to discuss any further questions.

---

### Official Review · Reviewer_n9xa · 2023-07-06

**Soundness:** 3 good
**Presentation:** 2 fair
**Contribution:** 3 good
**Rating:** 7
**Confidence:** 3

**Summary:**

The authors consider the problem of learning an adversarial low-rank infinite episodic MDP with unknown transition and full information feedback.
The idea behind this problem is that in many RL applications, the state and action spaces might be prohibitively large, rending results that scale with these metrics meaningless. Several approaches have used the assumption that there is a feature mapping state-action pairs to a low dimensional embedding space. While the common practice is to assume that this embedding is known, in the present problem the learner has to learn how the data is represented.
The second challenge faced by the learner is that the loss functions change adversarially. These two challenges have been studied separately, but this work is the first to study them simultaneously.

To do so, they work under the assumption that there exists an unknown low-rank embedding of the state-action pairs and propose an algorithm to tackle this problem: Policy Optimization for LOw rank MDPS (POLO). This algorithm relies on the standard Online Mirror Descent technique, combined with a bonus function which lowers the observed losses to favorize exploration.

The authors then provide a rigourous analysis of the regret and show that their algorithm reaches sublinear regret.
The regret analysis follows a standard  decomposition of the regret into an estimation bias term, the OMD regret term and an optimism term, which respectively compare the true value function with the eztimated value function of the policy played by the learner, the difference between the performance of the policy played by the learner and the best policy in hindsight on the estimated value function, and finally the  difference between the estimated value function and the true value function for the best policy.


**Strengths:**

This paper tackles a new and challenging problem by combining two notably difficult aspects of online learning with MDPs: handling adversarial losses and having to learn the state-action mapping. Each of these problems has received a lot of attention in the past few years and are really relevant for the online learning  community.

The authors achieve their goal by presenting an algorithm that adapts to both of these aspects simultaneously and obtains sublinear regret.
The algorithm that they present buil;dss upon standard tools in the litterature and has an appreciably clear and compact analysis.

The authors properly discuss the limitations of their work, notably by questioning the computational efficiency of their approach and ensuring that their algorithm is coomputationally feasible.

Overall, this paper will provide a good baseline for the problems of representation learning of MDPs in adversarial environments.


**Weaknesses:**

It would be interesting to provide a the lower bound for the problem: from the simple online learning with full information feedback, having at least $\sqrt{K}$ regret from playing adversarial losses is unavoidable.  In this work, it is stated that the first term in the regret bound, which scales with $K^{3/4}/(1 - \gamma)$ comes from learning the unknown transition kernel $P^\star$. Do you have a lower bound for this part the representation learning part of the problem?




**Questions:**

Is there a lower bound for this problem, and could you elaborate on the cost of learning the representation?

**Limitations:**

The authors provided an interesting discussion of the potential limitations of their work and addressed them adequately.

---

> ### Author Rebuttal · Authors · 2023-08-09
>
> We thank the reviewer for the valuable comments and suggestions. Our response to each question is provided in turn below.
>
> **Q1. "Is there a lower bound for this problem, and could you elaborate on the cost of learning the representation?"**
>
> We now provide a lower bound for the representation learning part of the problem, in the setting of learning low-rank MDPs with deterministic reward functions, which thus also serves as a regret lower bound for learning adversarial low-rank MDPs with full-information feedback. To the best of our knowledge, this is the first regret lower bound for low-rank MDPs. At a high level, we construct $dA$ hard MDP instances, which are difficult to distinguish in KL divergence but have very different optimal policies. Similar instances are first introduced to prove the regret lower bounds for tabular MDPs [1,2] and are recently used to prove the lower bound of sample complexity for learning low-rank MDPs by [3].
>
> **Theorem 1** (Regret Lower Bound)
>
> Suppose $d\geq 8$, $S\geq d+1$, $A\geq d-3$, and $K\geq 2(d-4)A$. Then for any algorithm $\operatorname{Alg}$, there exists an episodic infinite-horizon low-rank MDP $\mathcal{M}_{\operatorname{Alg}}$ with fixed reward function such that the expected regret for this MDP is lower bounded by $\Omega\left(\frac{\gamma^2}{1-\gamma} \sqrt{d A K}\right)$.
>
> **Remark 1**
> We note that this regret lower bound can hold when $d\ll S$ and $d\ll A$, which thus means that this lower bound is non-trivial. Besides, our Theorem 5.1 matches the regret lower bound in $A$ up to a logarithmic factor but looses a factor of $\widetilde{O}(K^{1/4}d^{1/2}/\gamma^2)$. More importantly, compared with the regret upper bound $\widetilde{O}\left(d \sqrt{ K/(1-\gamma)^3}\right)$ of linear MDPs [5] (the finite horizon $H$ is substituted by the effective horizon $\Theta(1/(1-\gamma))$), the dependence on $A$ in the regret lower bound of low-rank MDP shows a clear separation between low-rank MDPs and linear MDPs, which demonstrates that low-rank MDPs are statistically more difficult to learn than linear MDPs in the regret minimization setting.
>
> **Proof of Theorem 1**
> To begin with, we first construct a reference low-rank MDP $\mathcal{M}_0$, with its elements detailed as follows:
>
> * State space:
> $\mathcal{S}=\\{s_{1,1},s_{2,1},s_{2,2},\ldots,s_{2,d-4}\\}\cup\\{s_g,s_b\\}\cup\\{s_o\\}\cup\mathcal{S}_{\mathcal{O}}$,
> where
>
> $\mathcal{S}_{\mathcal{O}}= \{s_{o,i}\}_{i=1}^{S-d}$
>
> denotes the set of 'outlier states', $s_g$ denotes the 'good state', and $s_b$ denotes the 'bad state'.
> * Action space: $\mathcal{A}=\\{a_1,a_2,\ldots,a_{A}\\}$.
> * Reward function: $r(s,a)=\mathbb{I}\\{s=s_g\\}+\frac{1}{2}\mathbb{I}\\{s\in\mathcal{S}_{\mathcal{O}}\\}$.
> * Transitions:
>   * For the initial state $s_{1,1}$, the learner will deterministically transit to state $s_{2,i}$ if taking action $a_i$, $\forall i\in[d-4]$, and will transit to state $s_o$ otherwise. Formally, $P^\star\left(s_{2,i} \mid s_{1,1}, a_i\right)=1$, $\forall i\in[d-4]$, and $P^\star\left(s_o \mid s_{1,1}, a_i\right)=1$, $\forall i\in[A]\setminus[d-4]$.
>   * For state $s_{2,i}\in\\{s_{2,1},s_{2,2},\ldots,s_{2,d-4}\\}$, the learner will transit to good state $s_{g}$ and bad state $s_b$ uniformly at random, no matter what action it takes, $\textit{i.e.}$, $P^\star\left(s_{g} \mid s_{2,i}, a\right)=P^\star\left(s_{b} \mid s_{2,i}, a\right)=\frac{1}{2}$, $\forall i\in[d-4]$ and $a\in\mathcal{A}$.
>   * For states $s_o$ and $s_{o,i}\in\mathcal{S}_{\mathcal{O}}$, the learner will uniformly transit to a state
>
>  $s_{o,j}\in\mathcal{S}_{\mathcal{O}}$,
> no matter what action it takes.
>
> Formally, $P^\star\left(s_{o,j} \mid s_o, a\right)=P^\star\left(s_{o,j} \mid s_{o,i}, a\right)=\frac{1}{S-d}$, $\forall s_{o,i},s_{o,j}\in\mathcal{S}_{\mathcal{O}}$ and $a\in\mathcal{A}$.
>   * For states $s_g$ and $s_b$, the learner will stay at the current state no matter what action it takes, which means that $P^\star\left(s_{g} \mid s_g, a\right)=P^\star\left(s_{b} \mid s_{b}, a\right)=1$.
>
> Further, the transitions of the above MDP can be realized by $P^\star(s^\prime\mid s,a)=\langle\phi^\star(s,a),\mu^\star(s^\prime)\rangle$, with the following features, which thus implies that this MDP is indeed a low-rank MDP:
>
> $\mu^\star(s_{2,i})=\mathbf{e}_i,\forall i\in[d-4]$
>
> $\mu^\star(s_{1,1})=\mathbf{0}$
>
> $\mu^\star(s_o)=(0,\ldots, 0,1,0)$
>
> $\mu^\star(s_g)=(0, \ldots, 0,1,0,0,0)$
>
> $\mu^\star(s_b)=(0, \ldots, 0,0,1,0,0)$
>
> $\mu^\star(s_{o,j})=(0,\ldots,0,1)$
>
> $\phi^\star(s_{1,1},a_i)=\mathbf{e}_i,\forall i\in[d-4]$
>
> $\phi^\star(s_{1,1},a_i)=(0,\ldots, 0,1,0),\forall i\in[A]\setminus[d-4]$
>
> $\phi^\star\left(s_{2,j}, a\right)=(0, \ldots, 0, \frac{1}{2}, \frac{1}{2}, 0,0),\forall a\in \mathcal{A}$
>
> $\phi^\star(s_o,a)=\phi^\star(s_{o,j},a)=(0,\ldots,0,\frac{1}{S-d}),\forall a\in \mathcal{A}$
>
> $\phi^\star(s_g,a)=\mu^\star(s_g),\forall a\in \mathcal{A}$
>
> $\phi^\star(s_b,a)=\mu^\star(s_b),\forall a\in \mathcal{A}$
>
> Based on the reference MDP $\mathcal{M}_0$, we define other low-rank MDP instances
>
> $\mathcal{M}_{(i^\ast,a^\ast)}$, $\forall (i^\ast,a^\ast)\in[d-4]\times \mathcal{A}$.
> In specific, the only difference between
>
> $\mathcal{M}_{(i^\ast,a^\ast)}$
> and
> $\mathcal{M}_0$
>
> is that $\phi^\star(s_{2,i^\ast},a^\ast)=(0, \ldots, 0, \frac{1}{2}+\varepsilon, \frac{1}{2}-\varepsilon, 0,0)$, such that $P^\star\left(s_g \mid s_{2,i^\ast}, a^\ast\right)=\frac{1}{2}+\varepsilon$, $P^\star\left(s_b \mid s_{2,i^\ast}, a^\ast\right)=\frac{1}{2}-\varepsilon$, for some $\varepsilon>0$ to be defined later.
>
> In what follows, we denote by
> $\mathbb{P}_{\left(i^*, a^*\right)} \triangleq$
>
> $\mathbb{P}_{\operatorname{Alg},\mathcal{M}_{\left(i^*, a^*\right)}}$
>
> the probability measure over the outcomes induced by the interaction between
> $\operatorname{Alg}$ and
> $\mathcal{M}_{\left(i^*, a^*\right)}$, (please see the remaining proof in our 'global' response in the above)

---

> > ### Author Response · Authors · 2023-08-15
> > **About the equations that were not successfully rendered**
> >
> > We now rewrite the sentences containing the equations in the proof of Theorem 1 that were not successfully rendered:
> >
> > * where $\mathcal{S}\_{\mathcal{O}}=\\{s\_{o,i}\\}_{i=1}^{S-d}$ denotes the set of 'outlier states',
> > * we denote by $\mathbb{P}\_{\left(i^*, a^*\right)} \triangleq \mathbb{P}\_{\operatorname{Alg}, \mathcal{M}\_{\left(i^*, a^*\right)}}$ the probability measure over the outcomes induced by the interaction between $\operatorname{Alg}$ and $\mathcal{M}_{\left(i^*, a^*\right)}$,

---

> > > ### Comment · Reviewer_n9xa · 2023-08-16
> > >
> > > Thank you for providing this lower bound, and for providing detailed answers to all reviewers.
> > > I will adjust my score accordingly.

---

> > > > ### Author Response · Authors · 2023-08-16
> > > >
> > > > Thank you for raising your score for our work! We believe that our revised version will be strengthened with this new regret lower bound and we are happy to discuss any further questions.

---

### Official Review · Reviewer_qWwQ · 2023-07-10

**Soundness:** 3 good
**Presentation:** 3 good
**Contribution:** 2 fair
**Rating:** 5
**Confidence:** 3

**Summary:**

This work studies low-rank MDPs with unknown and fixed transition and full-information adversarial losses. The proposed algorithm generalize RepUCB from the fixed reward setting to the adversarial reward setting. The main idea of the algorithm is to replace the greedy policy in RepUCB to an incremental policy update with exponential weights. Also, the proposed algorithm interleaves exploration and exploitation to obtain low regret.

**Strengths:**

- This is an elegant generalization of RepUCB to the adversarial setting. It is also the first paper that simultaneously deals with non-linear function approximation and adversarial losses.

**Weaknesses:**

- The technical novelty is a bit limited -- Given OPPO and POWERS, it becomes more or less clear that the key technique for unknown transition + full-information loss feedback setting is to ensure optimism and use a no-regret algorithm over the optimistically-biased losses. Though the form of optimism in low-rank MDP (i.e., optimism only in the initial state) is slightly different from those in linear mixture MDP, with full-information loss feedback, the key challenges is already resolved in RepUCB. It is not surprising that most of the analysis follows those of RepUCB except for the OMD regret term.
- The proposed algorithm performs an epsilon-greedy styled exploration, which leads to a sub-optimal $K^{3/4}$ regret bound. To improve the technical novelty, I think the paper should try to design some less trivial exploration scheme and improve over the $K^{3/4}$ bound. Another direction to improve the technical novelty is to study the bandit feedback case.
- The comparison with Rep-UCB around Line 259-268 and the comment that your algorithm is tighter in $d, A, \gamma$ is slightly off. The reason is that your bound is better than RepUCB only when $K^{3/4}A^{1/2}d/(1-\gamma) < d^{4/3}A^{2/3}K^{2/3}/(1-\gamma)^{4/3}$, which is equivalent to $K<d^4A^2/(1-\gamma)^4$. In this regime, both regret bounds $K^{3/4}A^{1/2}d/(1-\gamma)$ and $d^{4/3}A^{2/3}K^{2/3}/(1-\gamma)^{4/3}$ are at least $K$, meaning that this is an *vacuous* regime where both regret bounds are meaningless. In other words, there is no case your regret bound is better than that of RepUCB. If we calculate the sample complexity, the two algorithms would give $\frac{A^2d^4}{(1-\gamma)^4\epsilon^4}$ and $\frac{A^2d^4}{(1-\gamma)^4\epsilon^3}$ respectively, so there is actually no improvement in terms of $d, A, (1-\gamma)$.

**Questions:**

See the above sections.

**Limitations:**

There is no societal limitation.

---

> ### Author Rebuttal · Authors · 2023-08-09
>
> We thank the reviewer for the valuable comments and suggestions. Our response to each question is provided in turn below.
>
> **Q1. "The technical novelty is a bit limited."**
>
> As previous works studying adversarial linear mixture MDPs using policy-optimization (PO) based methods [1,2] relying upon some ingredients of the analyses in stochastic linear mixture MDPs [3,4], our work also relies upon some ingredients of the analyses in stochastic low-rank MDPs [5]. Besides, we would also like to note that our work is not a straightforward combination of Rep-UCB and previous PO-based methods, and new ingredients are needed to achieve our regret guarantee, due to the technical issue which is unique to low-rank MDPs. In specific, as discussed in Section 4.1 and 5.1, the original Rep-UCB algorithm only has a sample complexity guarantee but does not enjoy a regret guarantee in both stochastic and adversarial environments, due to its uniform exploration in each episode. Moreover, a common ETC-style adaption of Rep-UCB also only attains an $\widetilde{O}(K^{2/3})$ regret in stochastic environments, but committing to a fixed loss function makes it still fail to be extended to adversarial environments. Therefore, directly applying previous PO-based methods [1,2] together with the current Rep-UCB algorithm does not go through our problem. Further, to tackle this issue, our algorithmic design features a new *doubled exploration and exploitation* scheme, which leverages a mixed roll-out policy to simultaneously conduct (a) the uniform exploration over transitions required by representation learning; and (b) the exploration and exploitation over adversarial loss functions. As discussed in Section 4.1 and 5.1, this is critical to achieving our final regret guarantee.
>
> Moreover, we now provide a regret lower bound for learning low-rank MDPs, which is the first regret lower bound for low-rank MDPs and together with our regret upper bound provides a more complete characterization of regret minimization for low-rank MDPs. Please see this in our response to reviewer n9xa.
>
> **Q2. "The proposed algorithm performs an epsilon-greedy styled exploration...To improve the technical novelty..."**
>
> Indeed, our doubled exploration and exploitation scheme is similar in spirit to the epsilon-greedy-styled exploration. However, we would also like to note that actually, they are inherently not the same, since the epsilon-greedy styled exploration will only conduct exploitation in certain rounds but our doubled exploration and exploitation scheme will always conduct exploration in all rounds, which will be either the exploration required by representation learning or the exploration (and exploitation) required to tackle the adversarial loss functions.
>
> As we discussed in Section 6, we do believe further optimizing the current dependence on $K$ and studying the bandit feedback case are interesting and important next steps. However, we would like to remark that our $\widetilde{O}(K^{3/4})$ regret guarantee obtained by the PO-based method is valuable and meaningful, especially considering that it is the first result for RL with both nonlinear function approximation and adversarial loss functions. In fact, the SOTA regret guarantee obtained by the PO-based method for the more amenable adversarial linear MDPs with given underlying feature mappings in the full-information setting is $\widetilde{O}(K^{3/4})$, as shown by a concurrent work [6]. Our algorithm obtains the regret guarantee with the same dependence on $K$ and can additionally work when no true feature mappings are known a priori. For the bandit feedback case, due to the local-search nature of PO-based methods, the current PO-based methods for adversarial linear MDPs [7,8,9] depend on the *dilated* exploration bonuses (and its variants), which critically rely on the given underlying true feature mappings, to facilitate global exploration. These PO-based methods are thus not applicable to our problem, with no true feature mappings given a priori. Moreover, we would like to note that, even with the true feature mappings given, learning adversarial linear MDPs with bandit feedback is already rather challenging, where the SOTA regret achieved by the PO-based method is only of $\widetilde{O}(K^{6/7})$ order [9] when no exploration conditions are given. Overall, learning adversarial low-rank MDPs with bandit feedback is indeed interesting and also particularly challenging, and we believe our work may serve as an important step to further tackle this problem, which we leave as our future study.
>
> **Q3. "The comparison ... is slightly off."**
>
> Thank you again for pointing this out. Though we mainly focus on learning low-rank MDPs in adversarial environments, the comparison between the regret guarantee of our algorithm and that of the ETC-style adaption of Rep-UCB in stochastic environments is indeed not accurate, and we will improve our statement on Line 264-266 accordingly in the revised version of our paper.
>
>
>
> [1] Cai et al. Provably Efficient Exploration in Policy Optimization. ICML, 20.
>
> [2] He et al. Near-optimal Policy Optimization Algorithms for Learning Adversarial Linear Mixture MDPs. AISTATS, 22.
>
> [3] Ayoub et al. Model-Based Reinforcement Learning with Value-Targeted Regression. ICML, 20.
>
> [4] Zhou et al. Nearly Minimax Optimal Reinforcement Learning for Linear Mixture Markov Decision Processes. COLT, 21.
>
> [5] Uehara et al. Representation learning for online and offline RL in low-rank mdps. ICLR, 22.
>
> [6] Zhong et al. A Theoretical Analysis of Optimistic Proximal Policy Optimization in Linear Markov Decision Processes. arXiv, 23.
>
> [7] Luo et al. Policy optimization in adversarial mdps: Improved exploration via dilated bonuses. NeurIPS, 21.
>
> [8] Dai et al. Refined Regret for Adversarial MDPs with Linear Function Approximation. ICML, 23.
>
> [9] Sherman et al. Improved Regret for Efficient Online Reinforcement Learning with Linear Function Approximation. ICML, 23.

---

> > ### Comment · Reviewer_qWwQ · 2023-08-16
> >
> > I agree that since the setting is not studied before, any result would be beneficial for future study, even though it's not close to optimal. I have raised my score.

---

> > > ### Author Response · Authors · 2023-08-16
> > >
> > > Thank you for your positive feedback and support for our work! We are also more than happy to answer any further questions.

---

### Author Rebuttal · Authors · 2023-08-09

**Proof of the Regret Lower Bound (Cont.)**
and by
$\mathbb{E}_{\left(i^*, a^*\right)} \triangleq$

$ \mathbb{E}_{\operatorname{Alg}, \mathcal{M}_{\left(i^*, a^*\right)}}$

the expectation with respect to $\mathbb{P}_{\left(i^*, a^*\right)}$.

**Step 1: Regret of $\operatorname{Alg}$ over $\mathcal{M}_{(i^\ast,a^\ast)}$**
For some $\mathcal{M}_{(i^\ast,a^\ast)}$,
its optimal policy

$\pi^\ast_{(i^\ast,a^\ast)}:\mathcal{S}\to\mathcal{A}$

satisfies that $\pi^\ast_{(i^\ast,a^\ast)}(s_{1,1})=a_{i^\ast}$ and $\pi^\ast_{(i^\ast,a^\ast)}(s_{2,i^\ast})=a^\ast$, with the optimal value function

$$
\begin{align}
V_{0}^*(s_{1,1}) & =\mathbb{E}\left[\sum_{\tau=0}^{+\infty} \gamma^\tau r(s_{\tau}, a_\tau) \mid \pi^\ast_{(i^\ast,a^\ast)},P^\star_{(i^\ast,a^\ast)}, s_0=s_{1,1}\right]=\sum_{\tau=2}^{+\infty} \gamma^\tau \left(\frac{1}{2}+\varepsilon\right)=\frac{\gamma^2}{1-\gamma}\left(\frac{1}{2}+\varepsilon\right).\tag{1}\label{eq:lb_eq1}
\end{align}
$$

For some policy $\pi$, it is also clear that its value function satisfies

$V_{0}^\pi(s_{1,1})$

$$
=\frac{\gamma^2}{1-\gamma}(\frac{1}{2}+\varepsilon\mathbb{P}_{(i^*, a^*)}((s_2,a_2)=(s_{2,i^*},a^*))).\tag{2}\label{eq:lb_eq2}
$$


Combining Eq. $\eqref{eq:lb_eq1}$ and Eq. $\eqref{eq:lb_eq2}$ shows that the regret of
$\operatorname{Alg}$ in $\mathcal{M}_{(i^\ast,a^\ast)}$
satisfies

$R_K(\operatorname{Alg}, \mathcal{M}_{(i^\ast,a^\ast}))$

$$
=\frac{\gamma^2\varepsilon}{1-\gamma}K\left(1-\frac{1}{K}\mathbb{E}_{(i^*,a^*)}\left[N^K_{(i^*,a^*)} \right]\right)\notag\,,
$$

where we define $N^K_{(i^\ast,a^\ast)}= \sum_{k=1}^K\mathbb{I}\{(s^k_2,a^k_2)=(s_{2,i^\ast},a^\ast)\}$.

**Step 2: Maximum regret of $\operatorname{Alg}$ over all possible $\mathcal{M}_{(i^\ast,a^\ast)}$**
With $R_K(\operatorname{Alg}, \mathcal{M}_{(i^\ast,a^\ast}))$ in the above equation, we can deduce that

$$
\begin{align}
    \max_{(i^\ast,a^\ast)}R_K(\operatorname{Alg},\mathcal{M}_{(i^\ast,a^\ast)})&\geq \frac{1}{(d-4)A}\sum_{(i^\ast,a^\ast)}R_K(\operatorname{Alg},\mathcal{M}_{(i^\ast,a^\ast}))\notag\\
    &\geq \frac{\gamma^2\varepsilon}{1-\gamma}K\left(1-\frac{1}{K(d-4)A}\sum_{(i^\ast,a^\ast)}\mathbb{E}_{(i^\ast,a^\ast)}\left[N^K_{(i^\ast,a^\ast)} \right]\right)\,.\tag{3}\label{eq:lb_eq3}
\end{align}
$$

To lower bound the above display, it remains to upper bound

$\sum_{(i^*,a^*)}\mathbb{E}_{(i^*,a^*)}[N^K_{(i^*,a^*)}]$.

To this end, by Lemma 1 in the work of [4] together with the fact that $N^K_{(i^\ast,a^\ast)}/K\in[0,1]$, it holds that

$$
\begin{align*}
    \operatorname{KL}\left(\operatorname{Ber}\left(\frac{1}{K}\mathbb{E}_0\left[N^K_{(i^\ast,a^\ast)}\right]\right),\operatorname{Ber}\left(\frac{1}{K}\mathbb{E}_{(i^\ast,a^\ast)}\left[N^K_{(i^\ast,a^\ast)}\right]\right)\right)\leq \operatorname{KL}\left(\mathbb{P}_0,\mathbb{P}_{(i^\ast,a^\ast)}\right)\,.
\end{align*}
$$


This implies that

$$
\begin{align*}
    \frac{1}{K}\mathbb{E}_{(i^\ast,a^\ast)}\left[N^K_{(i^\ast,a^\ast)}\right]
    &\leq \frac{1}{K}\mathbb{E}_0 \left[N^K_{(i^\ast,a^\ast)}\right]+\sqrt{\frac{1}{2}\operatorname{KL}\left(\mathbb{P}_0,\mathbb{P}_{(i^\ast,a^\ast)}\right)}\\
    &=\frac{1}{K}\mathbb{E}_0 \left[N^K_{(i^\ast,a^\ast)}\right]+\varepsilon\sqrt{2}\sqrt{\mathbb{E}_0\left[N^K_{(i^\ast,a^\ast)}\right]}\,,
\end{align*}
$$

where the inequality is due to Pinsker’s inequality that $(p-q)^2 \leq \frac{1}{2} \operatorname{KL}(\operatorname{Ber}(p), \operatorname{Ber}(q))$, for $p,q\in[0,1]$, and the equality comes from Lemma 15.1 of  [1] and Lemma 14 of [2] as well as assuming $0\leq\varepsilon\leq\frac{1}{4}$.

Based on this, one can see that

$$
\begin{align}
    \frac{1}{K}\sum_{(i^\ast,a^\ast)}\mathbb{E}_{(i^\ast,a^\ast)}\left[N^K_{(i^\ast,a^\ast)}\right]
    &\leq \frac{1}{K}\sum_{(i^\ast,a^\ast)}\mathbb{E}_0 \left[N^K_{(i^\ast,a^\ast)}\right]+\varepsilon\sqrt{2}\sum_{(i^\ast,a^\ast)}\sqrt{\mathbb{E}_0\left[N^K_{(i^\ast,a^\ast)}\right]}\notag\\
    &\leq 1+\varepsilon\sqrt{2}\sqrt{(d-4)AK}\,,\tag{4}\label{eq:lb_eq4}
\end{align}
$$

where the second inequality follows from using the Cauchy-Schwartz inequality together with the fact that $N^K_{(i^\ast,a^\ast)}\leq K$.

**Step 3: Optimizing $\varepsilon$ to lower bound the maximum regret** Substituting Eq. $\eqref{eq:lb_eq4}$ into Eq. $\eqref{eq:lb_eq3}$ leads to

$$
\begin{align*}
    \max_{(i^\ast,a^\ast)}R_K(\operatorname{Alg},\mathcal{M}_{(i^\ast,a^\ast)})
    &\geq\frac{\gamma^2\varepsilon}{1-\gamma}K\left(1-\frac{1}{(d-4)A}-\varepsilon\sqrt{2}\sqrt{\frac{K}{(d-4)A}}\right)\\
    &\geq\frac{1}{4\sqrt{2}}\cdot\frac{\gamma^2}{1-\gamma}\left(1-\frac{1}{(d-4)A}\right)^2\sqrt{(d-4)AK}\\
    &\geq \frac{361}{1600\sqrt{2}}\cdot\frac{\gamma^2}{1-\gamma}\sqrt{(d-4)AK}\,,
\end{align*}
$$
where the second inequality comes from by choosing $\varepsilon=\frac{1}{2\sqrt{2}}\left(1-\frac{1}{(d-4)A}\right)\sqrt{\frac{(d-4)A}{K}}$ and the last inequality is due to $d\geq 8$ and $A\geq d-3$. Finally, note that  $\varepsilon\leq \frac{1}{4}$ is guaranteed when $K\geq 2(d-4)A$. The proof is thus concluded.
**Q.E.D.**

[1] Lattimore et al. Bandit algorithms. Cambridge University Press, 20.

[2] Domingues et al. Episodic Reinforcement Learning in Finite MDPs: Minimax Lower Bounds Revisited. ALT, 21.

[3] Cheng et al. Improved Sample Complexity for Reward-free Reinforcement Learning under Low-rank MDPs. ICLR, 23.

[4] Garivier et al. Explore first, exploit next: The true shape of regret in bandit problems. Mathematics of Operations Research, 19.

[5] He et al. Nearly Minimax Optimal Reinforcement Learning for Linear Markov Decision Processes. ICML, 23.

---

> ### Author Response · Authors · 2023-08-15
> **Rewriting the Remaining Proof (Part 1)**
>
> Since some equations were not successfully rendered, we now rewrite the proof of this part.
>
> **Proof of the Regret Lower Bound (Cont.)**
>
> and by $\mathbb{E}\_{\left(i^*, a^*\right)} \triangleq \mathbb{E}\_{\operatorname{Alg}, \mathcal{M}\_{\left(i^*, a^*\right)}}$ the expectation with respect to $\mathbb{P}\_{\left(i^*, a^*\right)}$.
>
> **Step 1: Regret of $\operatorname{Alg}$ in $\mathcal{M}\_{(i^\ast,a^\ast)}$**
> For some $\mathcal{M}\_{(i^\ast,a^\ast)}$, its optimal policy $\pi^\ast_{(i^\ast,a^\ast)}:\mathcal{S}\to\mathcal{A}$ satisfies that $\pi^\ast\_{(i^\ast,a^\ast)}(s_{1,1})=a\_{i^\ast}$ and $\pi^\ast\_{(i^\ast,a^\ast)}(s\_{2,i^\ast})=a^\ast$, with the optimal value function
>
> $$
> \begin{align}
> V\_{0}^*(s\_{1,1}) & =\mathbb{E}\left[\sum_{\tau=0}^{+\infty} \gamma^\tau r(s\_{\tau}, a\_\tau) \mid \pi^\ast\_{(i^\ast,a^\ast)},P^\star\_{(i^\ast,a^\ast)}, s\_0=s\_{1,1}\right]=\sum\_{\tau=2}^{+\infty} \gamma^\tau \left(\frac{1}{2}+\varepsilon\right)=\frac{\gamma^2}{1-\gamma}\left(\frac{1}{2}+\varepsilon\right).\tag{1}
> \end{align}
> $$
>
> For some policy $\pi$, it is also clear that its value function satisfies
> $$
> \begin{align}
> V_{0}^\pi(s\_{1,1}) &=\frac{\gamma^2}{1-\gamma}\left(\frac{1}{2}+\varepsilon\mathbb{P}\_{\left(i^*, a^*\right)}\left((s\_2,a\_2)=(s\_{2,i^\ast},a^\ast)\right)\right).\tag{2}
> \end{align}
> $$
>
>
>
> Combining Eq. $(1)$ and Eq. $(2)$ shows that the regret of $\operatorname{Alg}$ in $\mathcal{M}\_{(i^\ast,a^\ast)}$ satisfies
> $$
> \begin{align*}
> R\_K(\operatorname{Alg}, \mathcal{M}\_{(i^\ast,a^\ast}))&=\frac{\gamma^2\varepsilon}{1-\gamma}K\left(1-\frac{1}{K}\mathbb{E}\_{(i^\ast,a^\ast)}\left[\sum_{k=1}^K\mathbb{I}\{(s^k\_2,a^k\_2)=(s\_{2,i^\ast},a^\ast)\} \right]\right)\\
>     &=\frac{\gamma^2\varepsilon}{1-\gamma}K\left(1-\frac{1}{K}\mathbb{E}\_{(i^\ast,a^\ast)}\left[N^K\_{(i^\ast,a^\ast)} \right]\right)\notag,
> \end{align*}
> $$
> where we define $N^K\_{(i^\ast,a^\ast)}= \sum\_{k=1}^K\mathbb{I}\\{(s^k\_2,a^k\_2)=(s\_{2,i^\ast},a^\ast)\\}$.
>
> **Step 2: Maximum regret of $\operatorname{Alg}$ over all possible $\mathcal{M}\_{(i^\ast,a^\ast)}$**
> With $R\_K(\operatorname{Alg}, \mathcal{M}\_{(i^\ast,a^\ast}))$ in the above equation, we can deduce that
> $$
> \begin{align}
>     \max\_{(i^\ast,a^\ast)}R\_K(\operatorname{Alg},\mathcal{M}\_{(i^\ast,a^\ast)})&\geq \frac{1}{(d-4)A}\sum\_{(i^\ast,a^\ast)}R\_K(\operatorname{Alg},\mathcal{M}\_{(i^\ast,a^\ast}))\notag\\
>     &\geq \frac{\gamma^2\varepsilon}{1-\gamma}K\left(1-\frac{1}{K(d-4)A}\sum\_{(i^\ast,a^\ast)}\mathbb{E}\_{(i^\ast,a^\ast)}\left[N^K\_{(i^\ast,a^\ast)} \right]\right).\tag{3}
> \end{align}
> $$
>
>
>
> To lower bound the above display, it remains to upper bound $\sum\_{(i^\ast,a^\ast)}\mathbb{E}\_{(i^\ast,a^\ast)}\left[N^K\_{(i^\ast,a^\ast)} \right]$. To this end, by Lemma 1 in the work of [4] together with the fact that $N^K\_{(i^\ast,a^\ast)}/K\in[0,1]$, it holds that
> $$
> \begin{align*}
>     \operatorname{KL}\left(\operatorname{Ber}\left(\frac{1}{K}\mathbb{E}\_0\left[N^K\_{(i^\ast,a^\ast)}\right]\right),\operatorname{Ber}\left(\frac{1}{K}\mathbb{E}\_{(i^\ast,a^\ast)}\left[N^K\_{(i^\ast,a^\ast)}\right]\right)\right)\leq \operatorname{KL}\left(\mathbb{P}\_0,\mathbb{P}\_{(i^\ast,a^\ast)}\right).
> \end{align*}
> $$
>
>
> This implies that
> $$
> \begin{align*}
>     \frac{1}{K}\mathbb{E}\_{(i^\ast,a^\ast)}\left[N^K\_{(i^\ast,a^\ast)}\right]
>     &\leq \frac{1}{K}\mathbb{E}\_0 \left[N^K\_{(i^\ast,a^\ast)}\right]+\sqrt{\frac{1}{2}\operatorname{KL}\left(\mathbb{P}\_0,\mathbb{P}\_{(i^\ast,a^\ast)}\right)}\\
>     &=\frac{1}{K}\mathbb{E}_0 \left[N^K\_{(i^\ast,a^\ast)}\right]+\varepsilon\sqrt{2}\sqrt{\mathbb{E}\_0\left[N^K\_{(i^\ast,a^\ast)}\right]},
> \end{align*}
> $$
>
> where the inequality is due to Pinsker’s inequality that $(p-q)^2 \leq \frac{1}{2} \operatorname{KL}(\operatorname{Ber}(p), \operatorname{Ber}(q))$, for $p,q\in[0,1]$, and the equality comes from Lemma 15.1 of  [1] and Lemma 14 of [2] as well as assuming $0\leq\varepsilon\leq\frac{1}{4}$.
>
> Based on this, one can see that
> $$
> \begin{align}
>     \frac{1}{K}\sum\_{(i^\ast,a^\ast)}\mathbb{E}\_{(i^\ast,a^\ast)}\left[N^K\_{(i^\ast,a^\ast)}\right]
>     &\leq \frac{1}{K}\sum\_{(i^\ast,a^\ast)}\mathbb{E}\_0 \left[N^K\_{(i^\ast,a^\ast)}\right]+\varepsilon\sqrt{2}\sum\_{(i^\ast,a^\ast)}\sqrt{\mathbb{E}\_0\left[N^K\_{(i^\ast,a^\ast)}\right]}\notag\\
>     &\leq 1+\varepsilon\sqrt{2}\sqrt{(d-4)AK},\tag{4}
> \end{align}
> $$
>
>
> where the second inequality follows from using the Cauchy-Schwartz inequality together with the fact that $N^K\_{(i^\ast,a^\ast)}\leq K$.

---

> ### Author Response · Authors · 2023-08-15
> **Rewriting the Remaining Proof (Part 2)**
>
> **Step 3: Optimizing $\varepsilon$ to lower bound the maximum regret** Substituting Eq. $(4)$ into Eq. $(3)$ leads to
> $$
> \begin{align*}
>     \max\_{(i^\ast,a^\ast)}R\_K(\operatorname{Alg},\mathcal{M}\_{(i^\ast,a^\ast)})
>     &\geq\frac{\gamma^2\varepsilon}{1-\gamma}K\left(1-\frac{1}{(d-4)A}-\varepsilon\sqrt{2}\sqrt{\frac{K}{(d-4)A}}\right)\\
>     &\geq\frac{1}{4\sqrt{2}}\cdot\frac{\gamma^2}{1-\gamma}\left(1-\frac{1}{(d-4)A}\right)^2\sqrt{(d-4)AK}\\
>     &\geq \frac{361}{1600\sqrt{2}}\cdot\frac{\gamma^2}{1-\gamma}\sqrt{(d-4)AK},
> \end{align*}
> $$
>
> where the second inequality comes from by choosing $\varepsilon=\frac{1}{2\sqrt{2}}\left(1-\frac{1}{(d-4)A}\right)\sqrt{\frac{(d-4)A}{K}}$ and the last inequality is due to $d\geq 8$ and $A\geq d-3$. Finally, note that  $\varepsilon\leq \frac{1}{4}$ is guaranteed when $K\geq 2(d-4)A$. The proof is thus concluded.
> **Q.E.D.**
>
> [1] Lattimore et al. Bandit algorithms. Cambridge University Press, 20.
>
> [2] Domingues et al. Episodic Reinforcement Learning in Finite MDPs: Minimax Lower Bounds Revisited. ALT, 21.
>
> [3] Cheng et al. Improved Sample Complexity for Reward-free Reinforcement Learning under Low-rank MDPs. ICLR, 23.
>
> [4] Garivier et al. Explore first, exploit next: The true shape of regret in bandit problems. Mathematics of Operations Research, 19.
>
> [5] He et al. Nearly Minimax Optimal Reinforcement Learning for Linear Markov Decision Processes. ICML, 23.

---

### Decision · Program_Chairs · 2023-09-21

**Decision:**

Accept (poster)

**Comment:**

The paper makes a great first algorithmic contribution in an interesting and practically relevant reinforcement learning setting.